# Selective transduction and photoinhibition of pre-Bötzinger complex neurons that project to the facial nucleus in rats affects nasofacial activity

**Mariana R Melo[1]\*, Alexander D Wykes[2,3], Angela A Connelly[1], Jaspreet K Bassi[1], Shane D Cheung[4], Stuart J McDougall[2], Clément Menuet[5], Ross AD Bathgate[2,6], Andrew M Allen[1,2]\***

[1]Department of Anatomy & Physiology, University of Melbourne, Melbourne, Australia; [2]Florey Institute of Neuroscience and Mental Health, Melbourne, Australia; [3]Florey Department of Neuroscience and Mental Health, University of Melbourne, Melbourne, Australia; [4]Biological Optical Microscopy Platform (BOMP) - University of Melbourne, Melbourne, Australia; [5]Institut de Neurobiologie de la Méditerrané, INMED UMR1249, INSERM, Aix-Marseille Université, Marseille, France; [6]Department of Biochemistry and Molecular Biology, University of Melbourne, Melbourne, Australia

**\*For correspondence:**
mariana.del@unimelb.edu.au (MRM);
a.allen@unimelb.edu.au (AMA)

**Abstract** The pre-Bötzinger complex (preBötC), a key primary generator of the inspiratory breathing rhythm, contains neurons that project directly to facial nucleus (7n) motoneurons to coordinate orofacial and nasofacial activity. To further understand the identity of 7n-projecting preBötC neurons, we used a combination of optogenetic viral transgenic approaches to demonstrate that selective photoinhibition of these neurons affects mystacial pad activity, with minimal effects on breathing. These effects are altered by the type of anesthetic employed and also between anesthetized and conscious states. The population of 7n-projecting preBötC neurons we transduced consisted of both excitatory and inhibitory neurons that also send collaterals to multiple brainstem nuclei involved with the regulation of autonomic activity. We show that modulation of subgroups of preBötC neurons, based on their axonal projections, is a useful strategy to improve our understanding of the mechanisms that coordinate and integrate breathing with different motor and physiological behaviors. This is of fundamental importance, given that abnormal respiratory modulation of autonomic activity and orofacial behaviors have been associated with the development and progression of diseases.

## Editor's evaluation

This important study advances our understanding of the composition and circuit organization of the preBötzinger complex (preBötC), the brainstem region that generates the respiratory rhythm and coordinates breathing with different motor and physiological behaviors in mammals. The authors present convincing evidence supporting their conclusion that within the preBötC region, there is a subgroup of output neurons that has axonal projections to the facial motor nucleus and provides respiratory-related modulation of nasofacial muscle activity, based on technically elegant, state-of-the-art combinatorial dual viral transgenic and optogenetic approaches in rats. This work will be of interest to neuroscientists and physiologists working on the neural control of breathing and other motor systems.

**eLife digest** While breathing seems to come easy, it is a complex process in which many muscles coordinate to allow air to flow into the lungs. These muscles also control the flow of air we breathe out to allow us to talk, sing, eat, or drink. The brain circuits that control these muscles, can also influence other parts of the brain.

The preBötzinger Complex, which is a key region of brainstem circuits that generate and control breathing, contains neurons that also project widely, connecting to other regions of the brain. This helps to modulate the sense of smell, emotional state, heart rate, and even blood pressure. Understanding how the preBötzinger Complex is organized can untangle how breathing can influence these other processes.

Melo et al. wanted to learn whether they could manipulate the activity of a subgroup of preBötzinger Complex neurons that project into the facial nucleus – a region of the brain that controls the muscles of the face when we breathe – without affecting breathing. If this can be done, it might also be possible to affect blood pressure by manipulating selective preBötzinger neurons, and thus the development of hypertension, without having any impact on breathing.

To test this hypothesis, Melo et al. used rats in which the activation of preBötzinger Complex neurons that project into the facial nucleus was blocked. This decreased the activity of the muscles around the nose with hardly any effect on breathing. Melo et al. also found that the state of consciousness of the rat (anesthetized or conscious) could affect how preBötzinger Complex neurons control these muscles.

Melo et al. also observed that preBötzinger Complex neurons projecting into the facial nucleus had projections into many other regions in the brainstem. This might help to the coordinate respiratory, cardiovascular, orofacial, and potentially other physiological functions.

The findings of Melo et al. set a technical foundation for exploring the influence of specific subgroups of preBötzinger Complex neurons on respiratory modulation of other physiological activities, including blood pressure and heart rate and in conditions, such as hypertension and heart failure. More broadly, most brain regions contain complex and heterogeneous groups of neurons and the strategy validated by Melo et. al. could be applied to unravel other brain-function relationships.

## Introduction

Breathing is a complex behavior requiring the coordination of multiple motor patterns in the face, upper airways, thorax, and abdomen to enable gas exchange between the external and internal environment and maintenance of blood gas homeostasis (*Del Negro et al., 2018*). Strict coordination of breathing with the activity of facial and upper airway muscles underlies the rhythmic orofacial behaviors, such as sniffing, sucking, licking, whisking, vocalization, and swallowing, which are fundamental for ingestive and exploratory behavior, and social interaction (*Deschênes et al., 2016*; *Deschênes et al., 2015*; *Huff et al., 2022*; *Moore et al., 2014*; *Takatoh et al., 2022*). Breathing-related information permeates the brain with respiratory modulation of suprapontine brain structures affecting cognitive function and emotional expression (*Ashhad et al., 2022*; *Lavretsky and Feldman PhD, 2021*; *Yackle et al., 2017*). Similarly, coordination of this motor activity with autonomic nervous system activity produces phasic fluctuations of blood pressure (BP) and heart rate (HR) that are essential for optimizing cardiac efficiency and organ blood flow (*Fisher et al., 2022*; *Menuet et al., 2020*; *O'Callaghan et al., 2020*; *Shanks et al., 2022*).

The network of neurons coordinating breathing is distributed throughout the brainstem, with a major group responsible for generating inspiration located in the pre-Bötzinger complex (preBötC) within the ventrolateral medulla oblongata (*Smith et al., 1991*). In rodents, the preBötC consists of a heterogeneous population of ~3000 neurons with distinct, but intermingled, excitatory and inhibitory subgroups (*Ashhad and Feldman, 2020*; *Del Negro et al., 2018*). Anatomical studies showed that preBötC sends parallel excitatory and inhibitory projections to target nuclei throughout the brain, including to regions responsible for respiratory motor activity (*Yang and Feldman, 2018*).

As indicated by their extensive projections, preBötC neurons modulate many circuits in addition to those required for breathing activity. For example, preBötC neurons project monosynaptically to key medullary centers for autonomic regulation, and modulate cardiorespiratory coupling (*Dempsey*

*et al., 2017*; *Menuet et al., 2020*; *Menuet et al., 2017*). Photoinhibition of preBötC neurons produces the expected apnea, but also causes a significant reduction of sympathetic vasomotor activity, reduces respiratory modulation of BP, and increases cardiac parasympathetic nerve activity to decrease HR and respiratory-sinus arrhythmia (*Menuet et al., 2020*). Neurons in the preBötC also modulate orofacial behaviors. Activation of the developing brain homeobox1 (Dbx1)-expressing preBötC neurons alters the networks responsible for the coordination of upper airways and swallowing, to decrease the probability of swallowing during inspiration (*Huff et al., 2022*). The preBötC also contributes to motor patterns that are necessary for exploratory behaviors, such as sniffing and whisking. In this case, a group of inhibitory preBötC neurons provide monosynaptic inputs onto the vibrissal premotor neurons in the intermediate reticular nucleus (vIRt) to facilitate synchronous whisking. In addition, preBötC contains facial premotor neurons that modulate nasal dilation and mystacial pad (MP) muscle activity to couple whisking to the breathing rhythm (*Deschênes et al., 2016*; *Takatoh et al., 2022*).

Understanding and identifying the preBötC subgroups that coordinate and integrate breathing with different motor and physiological behaviors is of fundamental importance, given that abnormal respiratory modulation of autonomic activity and orofacial behaviors has been associated with the development and progression of diseases (*El-Omar et al., 2001*; *Huff et al., 2022*; *Menuet et al., 2017*; *Simms et al., 2009*). Although several studies have attempted to understand the physiological function/s of each preBötC neuronal subpopulation, disturbances in breathing, that occur when preBötC neurons are either inhibited or activated, impact our understanding of whether the observed changes are due to the direct effects of preBötC neurons or alterations resulting from changes in the respiratory rhythm and blood gases (*Cui et al., 2016*; *Huff et al., 2022*; *Menuet et al., 2020*; *Tan et al., 2008*; *Yackle et al., 2017*). Furthermore, whilst substantial gains have been made in understanding the expression of selective molecular markers in subgroups of preBötC neurons, it is rarely possible to reliably assign particular neurochemical signatures to unique functions (*Ashhad and Feldman, 2020*; *Del Negro et al., 2018*; *Yackle et al., 2017*). For example, while Dbx1+somatostatin (SST)+ predominantly affect the breathing pattern (*Cui et al., 2016*; *Del Negro et al., 2018*; *Ashhad and Feldman, 2020*), some Dbx1+SST+ which also express the neurokinin-1 receptor (NK1R) and, in some cases, cadherin-9 (Cdh9+), promote generalized behavioral arousal (*Yackle et al., 2017*).

To further understand the organization of the preBötC, in this study, we have tested the hypothesis that the preBötC region is composed of segregated subgroups of output neurons, potentially driven by a separate group of rhythmogenic neurons, that modulate specific behaviors, such as orofacial muscle activity. If correct, we predict that selective photoinhibition of preBötC neurons projecting to the facial nucleus (7n) should modulate orofacial behavior with minimal impact on respiratory motor activity. To test this hypothesis, we utilized a combinatorial viral transgenic approach with one virus providing cre-recombinase (Cre)-dependent expression of an optically activated chloride channel, GtACR2, and another retrograde axonal delivery of Cre. We showed that selective transduction and photoinhibition of preBötC facial premotor neurons affects the MP activity while minimally affecting breathing.

## Results

### Identification of different subpopulations of preBötC neurons based on their anatomical projection

To evaluate whether distinct subgroups of preBötC neurons could be distinguished based on their axonal projections, retrograde adeno-associated viruses (AAVrg), expressing either green fluorescent protein (GFP), or the red fluorophore, mCherry, were injected into 7n and the rostral ventrolateral medulla (RVLM) (n=3), respectively (*Figure 1A*). Three weeks later, fluorophore expression was observed in preBötC neurons, with overlapping distribution (*Figure 1B*), albeit with the RVLM projecting cells tending to be more medial. Most cells only contained a single fluorophore, but a small number of double-labeled neurons were also identified (*Figure 1C*). The injection sites were also confirmed by local transduction into the 7n (*Figure 1D*) and RVLM (*Figure 1E*). These results provide confidence that subgroups of preBötC neurons can be transduced based on their anatomical connection.

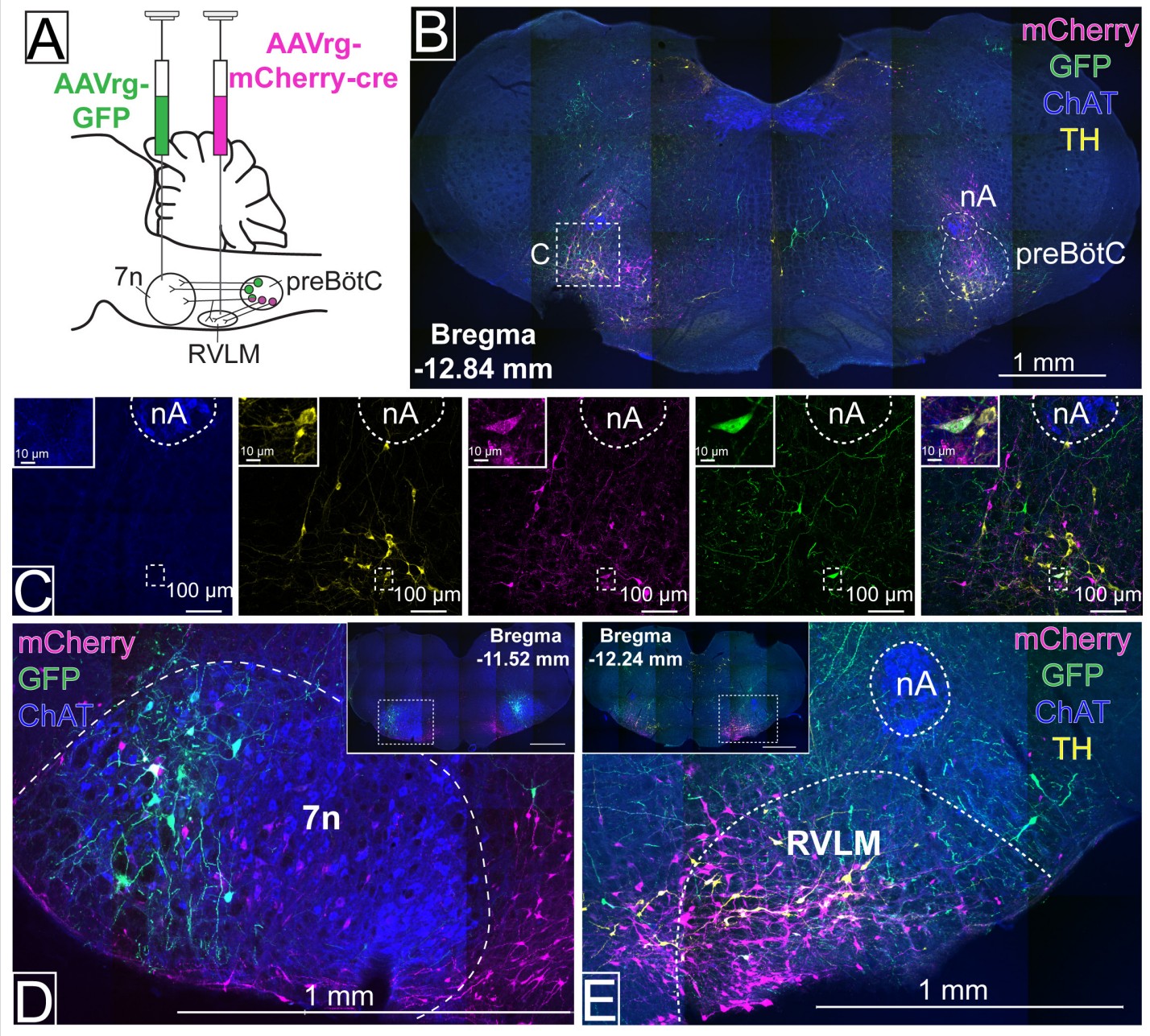

**Figure 1.** Distinct pre-Bötzinger complex (preBötC) subpopulations project to the facial nucleus (7n) and rostral ventrolateral medulla (RVLM). (**A**) Schematic diagram showing the dual retrograde viral strategy. (**B**) Coronal section showing preBötC neurons projecting to 7n (green) and RVLM (red). Tyrosine hydroxylase (TH; yellow) and choline acetyltransferase (ChAT; blue) neurons are shown. (**C**) Transduced preBötC neurons, highlighted in the hashed box in (**B**), shown in higher magnification. The inset shows a preBötC neuron that projects to both 7n and RVLM. Local transduction shows the 7n (**D**) and RVLM (**E**) injection sites. Abbreviations: nA: nucleus ambiguus.

## Inhibition of preBötC neurons affects multiple physiological functions in urethane-anesthetized rats

To modulate the activity of selectively transduced neurons, we generated an adeno-associated virus (AAV) which enabled Cre-dependent (DIO) expression, under the control of the ubiquitous chicken β-actin/cytomegalovirus (CAG) promoter, of the light-activated chloride channel, GtACR2, fused to a portion of $K_v2.1$ to promote predominant somatic targeting (*Lim et al., 2000*), and an enhanced GFP (MuGFP; *Scott et al., 2018*); hereafter called AAV-DIO-GtACR2-MuGFP. This was co-injected with a Cre-expressing virus, AAVDJ8-Cre-TdTomato, in male Sprague-Dawley (SD) rats (n=12) (*Figure 2A*),

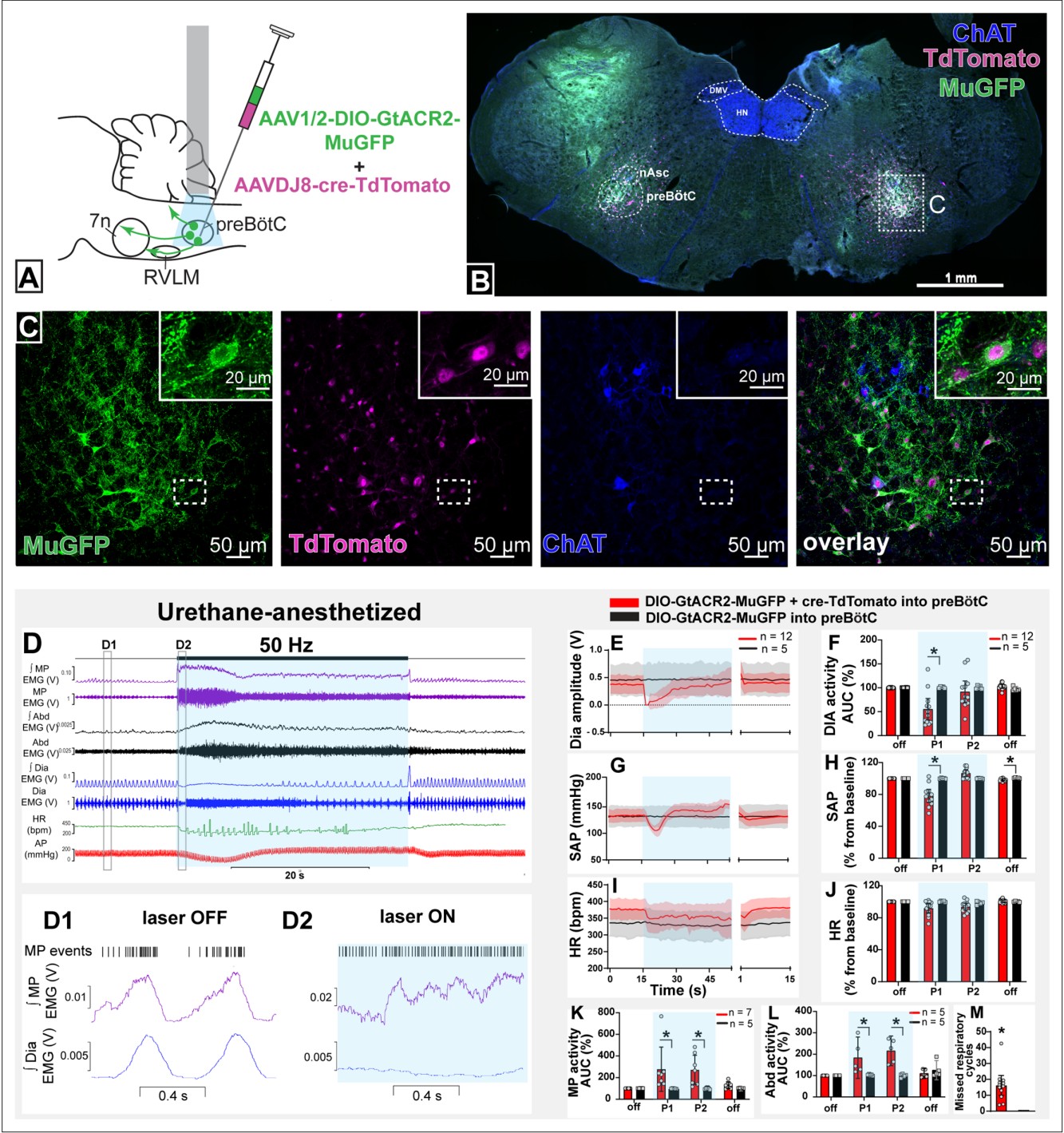

**Figure 2.** Effect on non-selective photoinhibition of pre-Bötzinger complex (preBötC) on cardiovascular, respiratory, and mystacial pad (MP) activity in urethane-anesthetized rats. (**A**) Schematic diagram showing the preBötC injection strategy. (**B**) Coronal section showing the expression of GtACR2-MuGFP and TdTomato in the preBötC, and choline acetyltransferase immunoreactivity (ChAT). Detailed maps showing the distribution of GtACR2-MuGFP expression are in *Figure 2—figure supplement 1*. (**C**) Higher magnification confocal images of MuGFP and TdTomato in preBötC with the inset highlighting a double-labeled neuron. (**D**) Representative traces showing recordings of integrated (∫) and raw MP EMG, ∫ and raw abdominal muscle (Abd) EMG, ∫ and raw diaphragm (Dia) EMG, heart rate (HR), and arterial pressure (AP). Bilateral photoinhibition of preBötC, which occurs in the time highlighted by a blue box, produced increased, tonic MP and Abd activity, apnea, a biphasic change in AP, and decreased HR. Higher temporal resolution recordings of the periods highlighted by the hashed boxes are shown in (**D1**) and (**D2**), which show MP bursts as events, and triggered averages of ∫ MP and ∫ Dia EMG. Note that the inspiratory-related MP activity was interrupted by photoinhibition and MP EMG became increased and tonic. Group data showing mean (solid line) and 95% confidence intervals for (**E**) Dia amplitude, (**G**) systolic arterial pressure (SAP), (**I**) HR (bpm) before,

*Figure 2 continued on next page*

*Figure 2 continued*

during, and after photoinhibition in GtACR2 expressing (red) and control (black) rats. Histograms showing group data for the effect of photoinhibition of preBötC in respiratory and cardiovascular parameters (**F**) Dia amplitude, (**H**) SAP, (**J**) HR, (**K**) MP activity, (**L**) Abd activity, and (**M**) number of missed respiratory cycles; P1 and P2 refer to the initial period of photoinhibition, where apnea was complete, and the later period where breathing re-started respectively. The group data are presented as mean ± 95% CI; unpaired t-test or nonparametric Mann-Whitney test with multiple comparisons using the Bonferroni-Dunn method, *p<0.05. Abbreviations: nAsc: subcompact formation of the nucleus ambiguus; DMV: dorsal motor nucleus of the vagus; HN: hypoglossal nucleus. Control injections of AAV-DIO-GtACR2-MuGFP alone into preBötC are shown in *Figure 2—figure supplement 2*.

The online version of this article includes the following source data and figure supplement(s) for figure 2:

**Source data 1.** Source data and statistics for *Figure 2*.

**Figure supplement 1.** Expression of GtACR2-MuGFP in rats co-injected with AAV-DIO-GtACR2-MuGFP and AAVDJ8-Cre-TdTomato in the pre-Bötzinger complex (preBötC).

**Figure supplement 2.** Control experiments with injection of AAV-DIO-GtACR2-MuGFP alone into pre-Bötzinger complex (preBötC).

to transduce the preBötC neurons in the region ventral and caudal to the tip of compact nucleus ambiguus. Immunohistochemical analysis showed that Cre recombination was effective and resulted in GtACR2-MuGFP expression in the preBötC (*Figure 2B*; *Figure 2—figure supplement 1A*). The $K_v2.1$ motif improved the neuronal expression of GtACR2, compared to GtACR2 alone (*Menuet et al., 2020*), with a clear definition of neuronal boundaries (*Figure 2C*). However, as previously demonstrated by other studies, the incorporation of the Kv2.1 motif into our viral construction did not prevent axonal expression of GtACR2-MuGFP (*Mahn et al., 2018*; *Messier et al., 2018*). At the caudal extent, a small proportion of GtACR2-MuGFP neurons also expressed parvalbumin, indicating likely transduction of some bulbospinal neurons of the rostral ventral respiratory group (rVRG; *Figure 2—figure supplement 1B*) and some parvalbumin vIRt neurons, that are located medially to nucleus ambiguus (*Figure 2—figure supplement 1C–F*). Control injections of AAV-DIO-GtACR2-MuGFP alone into preBötC did not produce any transduction (*Figure 2—figure supplement 2A–C*). Photoinhibition of these non-selectively transduced preBötC neurons in urethane-anesthetized rats produced very similar results to those described previously using a non-Cre-dependent approach (*Menuet et al., 2020*). This included immediate interruption of breathing and long-lasting apnea. Apnea did not last for the entire photoinhibition period, with inspiration resuming 9.3 s (95% CI: 6.3–12.3 s), after the beginning of the stimulus, albeit with a slower frequency and shorter diaphragm electromyograpy (dEMG) amplitude during continued photoinhibition (*Figure 2D–F and M*). We also observed a biphasic BP response, with a depressor response during the apnea and a pressor response upon resumption of breathing (*Figure 2D, G, and H*). We also observed a reduction in HR that lasted the entire period of photoinhibition, although it was not statistically different (*Figure 2I and J*).

Orofacial muscle activity, measured from electrodes inserted into the MP near the rostral tip of the snout - the site of origin of the muscle *nasolabialis profundus* (*Haidarliu et al., 2010*) - showed activity during the pre-inspiratory phase of the respiratory cycle, associated with the initial phase of vibrissae protraction (*Deschênes et al., 2016*). With non-selective photoinhibition of preBötC neurons, the MP activity increased in amplitude and shifted and became tonic (*Figure 2D, D1 and D2 and K*) with loss of inspiratory-related modulation. The onset of these effects correlated with apnea. When breathing resumed, during continued photoinhibition, the inspiratory-related MP activity also recovered, with increased amplitude, for the remainder of the stimulation period. Photoinhibition of the preBötC also immediately increased and produced tonic activity of the abdominal muscle that lasted for the apnea duration (*Figure 2D and L*). In some rats, rhythmic expiratory activity was observed with the return of dEMG inspiratory activity. Our results show that preBötC exerts a complex influence on orofacial activity as well as the timing of active expiration, potentially via the parafacial respiratory group.

No changes in any of these parameters were observed in control rats injected with only AAV-DIO-GtACR2-MuGFP (n=5) (*Figure 2E–M* and *Figure 2—figure supplement 2D*).

## Photoinhibition of preBötC neurons that project to the facial nucleus

We injected AAVrg-mCherry-Cre into the lateral and dorsal lateral border of the facial nucleus (n=6). This region is known to contain respiratory-modulated facial motoneurons that receive projections from the preBötC (*Deschênes et al., 2016*; *Takatoh et al., 2013*), with the lateral most edge of the facial nucleus containing motoneurons that regulate pre-inspiratory phase activity of the facial *nasolabialis profundus* muscle to cause vibrissae protraction (*Deschênes et al., 2016*). In the same surgery,

we also injected AAV-DIO-GtACR2-MuGFP into the preBötC with the aim of selectively transducing preBötC neurons projecting to the facial nucleus (preBötC→7n; *Figure 3A*).

This combinatorial approach resulted in GtACR2-MuGFP expression in a restricted, defined subgroup of preBötC neurons mostly located ventral to the extension of the subcompact nucleus ambiguus (*Figure 3B and C* and *Figure 3—figure supplement 1A*). Some of the transduced neurons were also found intermingled with parvalbumin-expressing rVRG neurons, but no co-localization between MuGFP and parvalbumin was observed (*Figure 3—figure supplement 1B*). A very small number of non-parvalbumin neurons (vIRt$_{PV}$), as well as sparse axonal labeling, was also observed medial to the nucleus ambiguus, in the region of the vIRt (*Figure 3—figure supplement 1C–F*).

In urethane-anesthetized rats, photoinhibition of preBötC→7n neurons significantly reduced the amplitude of the MP activity for 7.3 s (95% CI: 0.5–14.2 s), with the complete abolition of the inspiratory-related activity of the MP in half of the rats (MP activity baseline: 26.6 mV [95% CI: 9.4–43.8 mV] vs. during photoinhibition: 3.9 mV [95% CI: 1–6.8 mV], p=0.02) (*Figure 3D, E, and L*). In some rats, we observed a small decrease in dEMG amplitude (baseline: 29.1 mV [95% CI: 11.2–47 mV] vs. during photoinhibition: 23.8 mV [95% CI: 9.9–37.5 mV], p=0.04; *Figure 3D, E, G, M, and N*).

A slight reduction in BP was also apparent at the beginning of the photoinhibition (*Figure 3H, I*). Photoinhibition of preBötC→7n did not affect HR or Abd activity (*Figure 3J, K, and M*). These results clearly identify a subgroup of preBötC neurons providing inspiratory modulation of facial motoneurons that innervate extrinsic protractor muscles of the MP, which appear to be largely independent of those that drive inspiratory activity to the diaphragm, abdominal muscles, and autonomic nervous system.

## Anatomical and neurochemical characterization of transduced preBötC neurons

To characterize the anatomical distribution and neurochemical phenotype of preBötC→7n neurons, we combined immunohistochemistry and RNAscope, to identify mRNA expression. Following non-selective transduction of preBötC, GtACR2-MuGFP expression occurred between 12.5 and 13.92 mm caudal to Bregma (*Figure 4A and B*). We examined expression in detail in three rats and counted 1027 transduced neurons [95% CI: 769–1286 neurons] in each rat. Of these, 60.8% [95% CI: 59.2–62.4%] neurons expressed the vesicular inhibitory amino acid transporter (VIAAT, also called VGAT), and 32.3% [95% CI: 30.5–34.1] the vesicular-glutamate transporter 2 (VGlut2; *Figure 4B–D*). Further neuro-chemical characterization revealed that some MuGFP-expressing neurons also express somatostatin (SST) and/or reelin, and some NK1R (*Figure 4E–F*). The preBötC→7n transduced neurons were more restricted and located more caudally, between 13 and 13.76 mm caudal to Bregma (*Figure 5A and B*). In three rats, we counted 97 transduced neurons (95% CI: 32–162 neurons) in each rat. We found that 48.5% (95% CI: 12.77–84.23%) of muGFP neurons also expressed mCherry. We recognize that the percentage of co-localization seems to be lower than expected. However, considering that the Cre-loxP system requires only a single molecule of Cre-recombinase enzyme (*Van Duyne, 2015*), it is plausible that low levels of Cre, that were not detected by immunohistochemistry, had resulted in the expression of muGFP. As expected from previous reports (*Yang and Feldman, 2018*), a similar proportion of transduced preBötC→7n neurons expressed VGAT 38.6% (95% CI: 35.6–41.6%) or VGlut2 38% (95% CI: –7% to 82.8%) (*Figure 5B–D*). We also found that some preBötC→7n transduced neurons also co-express SST, reelin, or NK1R (*Figure 5E–G*).

## Distribution of the axonal projections of preBötC neurons projecting to the facial nucleus

Photoinhibition of preBötC→7n neurons produced a small effect on breathing and BP, although not statistically significant across the cohort. As suggested by our initial retrograde labeling experiments, where we observed a small number of preBötC with axon collaterals to both 7n and RVLM, we wondered whether collateralization might explain the small effects of photoinhibition of preBötC→7n neurons on non-facial motor outputs. Non-selective transduction of preBötC neurons resulted in a dense GtACR2-MuGFP-axonal expression in multiple brainstem nuclei (*Figure 6*), including the nucleus ambiguus (*Figure 6B*), RVLM (*Figure 6C*), Bötzinger complex (BötC) (*Figure 6C*), hypoglossal nucleus (*Figure 6D*), nucleus of the solitary tract (NTS) (*Figure 6F*), 7n (*Figure 6E*), rVRG (*Figure 2—figure supplement 1B*), and vIRt (*Figure 2—figure supplement 2B–E*).

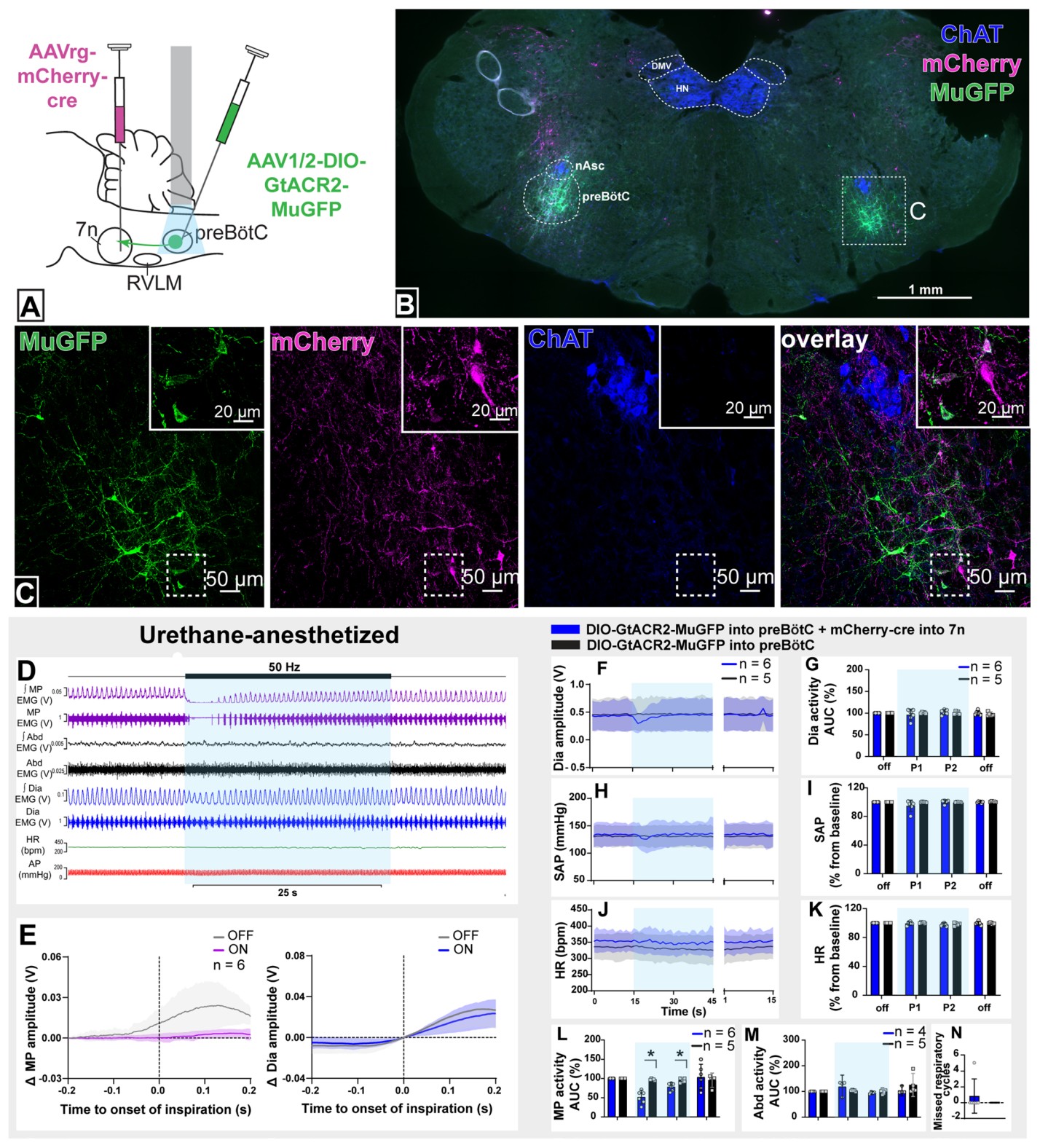

**Figure 3.** Effect of selective inhibition of pre-Bötzinger complex (preBötC)→7n neurons on cardiovascular, respiratory, and nasofacial activity in urethane-anesthetized rats. (**A**) Schematic diagram showing the injection protocol for selective transduction of preBötC→7n neurons. (**B**) Coronal section showing the expression of GtACR2-MuGFP and mCherry into the preBötC and immunohistochemistry for choline acetyltransferase (ChAT). Detailed maps showing the distribution of the expression of GtACR2-MuGFP are shown in *Figure 3—figure supplement 1*. (**C**) Higher magnification confocal image showing the co-localization of MuGFP with mCherry in neurons of preBötC. (**D**) Representative trace showing the integrated (∫) and raw

*Figure 3 continued on next page*

*Figure 3 continued*

mystacial pad (MP) EMG, ∫ and raw abdominal muscle (Abd) EMG, ∫ and raw diaphragm (Dia) EMG, heart rate (HR), and arterial pressure (AP). Bilateral selective photoinhibition of preBötC→7n neurons, highlighted with the blue box, decreased the overall activity, and interrupted the inspiratory-related MP activity, with minimal effects on respiratory, cardiovascular, or abdominal muscle activity. (**E**) Event-triggered average showing the magnitude of inspiratory-related modulation of MP activity in the 150–200 ms before and after the onset of inspiration - denoted by the dotted line at time 0. Note that the inspiratory-related MP activity ceased, even in the absence of the interruption of inspiratory activity. Group data showing mean (solid line) and 95% confidence intervals for (**F**) Dia amplitude, (**H**) systolic arterial pressure (SAP), and (**J**) HR (bpm) before, during, and after photoinhibition in selective GtACR2 expressing (blue) and control (black) rats. Histograms showing group data for the effect of photoinhibition of preBötC on respiratory and cardiovascular parameters (**G**) Dia amplitude, (**I**) SAP, (**K**) HR, (**L**) MP activity, (**M**) Abd activity, and (**N**) number of missed respiratory cycles; P1 and P2 refer to the initial period of photoinhibition, where there is a small decrease in Dia amplitude and the later period respectively. Group data are presented as mean ± 95% CI; unpaired t-test or nonparametric Mann-Whitney test with multiple comparisons using the Bonferroni-Dunn method, *p<0.05. The period of photoinhibition of preBötC neurons is depicted by blue shading. Abbreviations: DMV: dorsal motor nucleus of the vagus; nAsc: subcompact formation of the nucleus ambiguus; HN: hypoglossal nucleus.

The online version of this article includes the following source data and figure supplement(s) for figure 3:

**Source data 1.** Source data and statistics for *Figure 3*.

**Figure supplement 1.** Expression of GtACR2-MuGFP in selective pre-Bötzinger complex (preBötC)→7n transduced rats.

Strong axonal labeling was observed in the 7n, particularly along the dorsal and lateral edge of the nucleus following transduction of preBötC→7n neurons (*Figure 7*). In these animals, we also observed sparse GtACR2-MuGFP-axonal expression in the nucleus ambiguus (*Figure 7B*), BötC (*Figure 7C*), RVLM (*Figure 7C*), hypoglossal nucleus (*Figure 7D*), NTS (*Figure 7F*), rVRG (*Figure 2—figure supplement 1D*), and vlRt (*Figure 2—figure supplement 2F–I*).

These results indicate widespread collateralization of preBötC neurons into distinct areas of the brainstem. Whether this is the result of single neurons projecting to many targets or many neurons projecting to a small number of targets remains to be investigated. We speculate that this anatomical organization may contribute to the complex synchronization between respiratory, cardiovascular, orofacial, and potentially other physiological functions.

## Selective inhibition of preBötC neurons projecting to 7n in conscious rats

The rhythmic orofacial activities, such as whisking and sniffing, require a hierarchical organization of brainstem circuitry that involves the whisking premotor neurons in the vlRt and the preBötC respiratory-related premotor neurons (*Deschênes et al., 2016*; *Takatoh et al., 2022*). The activity of the whisking premotor neurons in vlRt is strongly affected by anesthesia and animal state (*Deschênes et al., 2016*; *Deschênes et al., 2015*). Here, we evaluated whether the response to photoinhibition of preBötC neurons was also affected by the type of anesthesia and the animals' state. Non-selective preBötC (n=5) and selective preBötC→7n (n=5) photoinhibition were performed under surgical ketamine/medetomidine anesthesia, during initial recovery to consciousness after reversal of anesthesia with atipamezole (1 mg/kg, i.p.), and 1–2 hr after reversal of anesthesia.

### Surgical ketamine/medetomidine anesthesia

In contrast to the urethane-anesthetized rats, during surgical ketamine/medetomidine anesthesia the MP activity was minimal and non-rhythmic (*Figure 8—figure supplement 1B, L*; *Figure 8—figure supplement 2A*, *Figure 8—figure supplement 3A*). Non-selective photoinhibition of preBötC neurons induced long-lasting apnea without affecting MP activity or HR (*Figure 8—figure supplement 1A–J*, *Figure 8—figure supplement 2A*). Photoinhibition of preBötC→7n neurons produced minimal effects on any parameter (*Figure 8—figure supplement 1K–T*). In one out of five rats, we observed a short apnea (3.9 s) (*Figure 8—figure supplement 1S*).

### Early recovery phase

During this phase, respiratory rate and HR gradually increased, and the inspiratory modulation of the MP activity resumed (*Figure 8B and L*, *Figure 8—figure supplement 2B*, *Figure 8—figure supplement 3B*). Spontaneous opening of the nares and whisker motion were visualized. Photoinhibition of preBötC immediately induced apnea, with MP activity becoming tonic, with no obvious inspiratory

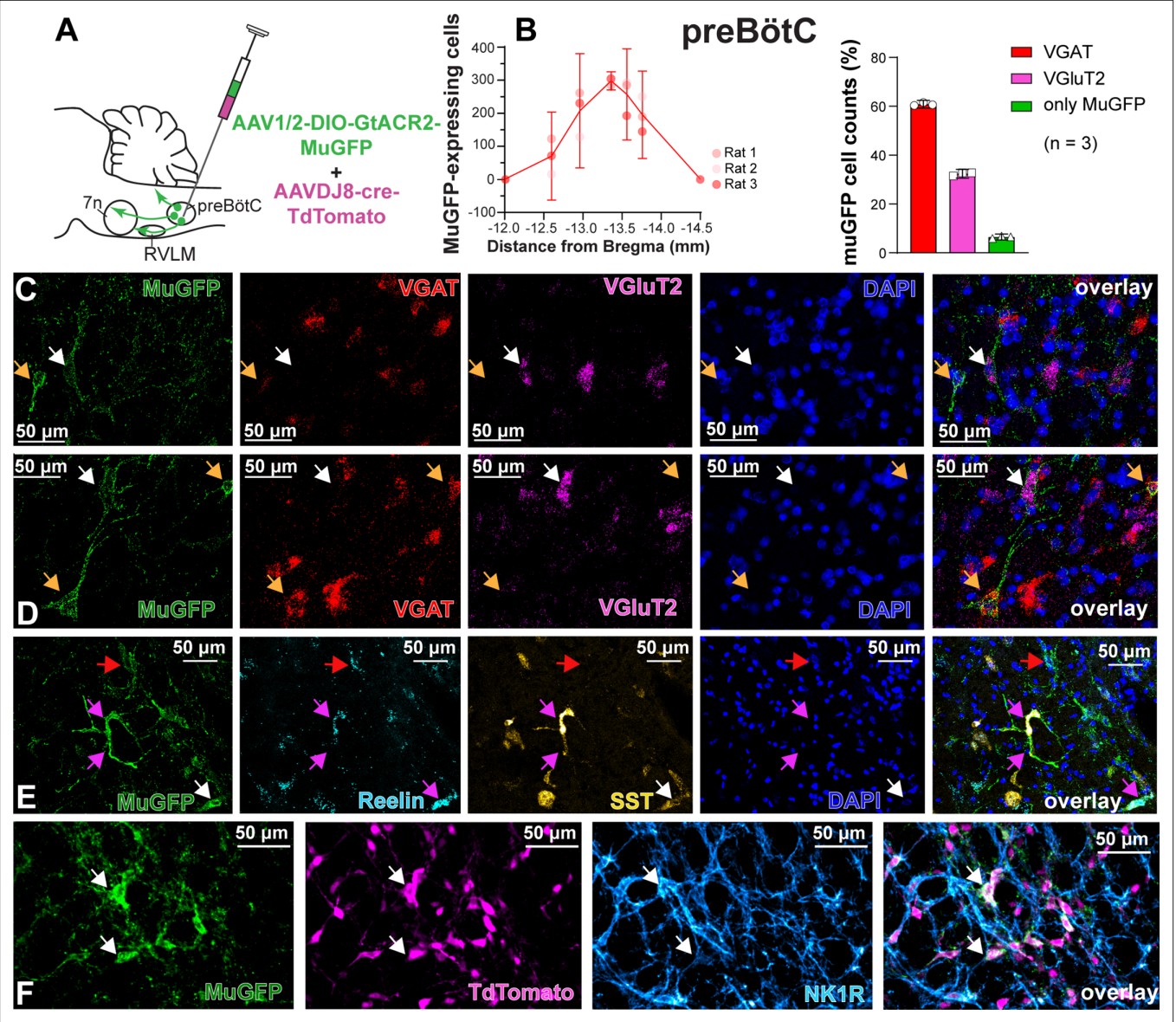

**Figure 4.** Excitatory and inhibitory pre-Bötzinger complex (preBötC) neurons are transduced by non-selective transfection of preBötC neurons. (**A**) Schematic diagram showing injection protocol for non-selective transduction of preBötC neurons. (**B**) Quantification of the total number of MuGFP-expressing neurons, plotted as the distance from Bregma (mm). The histograms show the number of transduced cells that co-expressed mRNA for *VGAT* or *VGluT2*. The results are presented as mean ± 95% CI. (**C–E**) In situ hybridization showing the co-expression of MuGFP (green), with mRNA for VGAT (red), VGluT2 (magenta), reelin (cyan), and somatostatin (SST) (yellow) in preBötC. Nuclei are labeled in blue (DAPI). The yellow arrows highlight *VGAT* neurons, the white arrows highlight *VGluT2* neurons, the red arrows highlight reelin neurons, and the pink arrows highlight reelin and SST neurons. (**F**) Immunohistochemistry showing that some neurons expressing MuGFP (green) and TdTomato (magenta) also express neurokinin-1 receptor (NK1R) (light blue). Co-localization is indicated by white arrows.

modulation (*Figure 8A–F and I*, *Figure 8—figure supplement 2B*). The amplitude of the MP activity increased in all five rats, but this was not statistically significant (*Figure 8J*). With the resumption of breathing, MP activity returned to baseline, even when this occurred during continued light delivery. The HR was not affected (*Figure 8G and H*). Selective inhibition of preBötC→7n neurons did not induce apnea, although a slight reduction in respiratory frequency occurred (*Figure 8K–P and S*). In contrast to observations under the urethane anesthesia, photoinhibition of preBötC→7n neurons increased the overall MP activity during the first 5.1 s after the onset of the stimulus (95% CI: 2.9–7.4 s; *Figure 8L and T*) and interrupted the inspiratory-related modulation in three out of five rats (1 mV [95% CI: 0.09–1.9 mV] vs. baseline: 6.8 mV [95% CI: 6.4–7.8 mV], p=0.01). An increase in respiratory-related MP

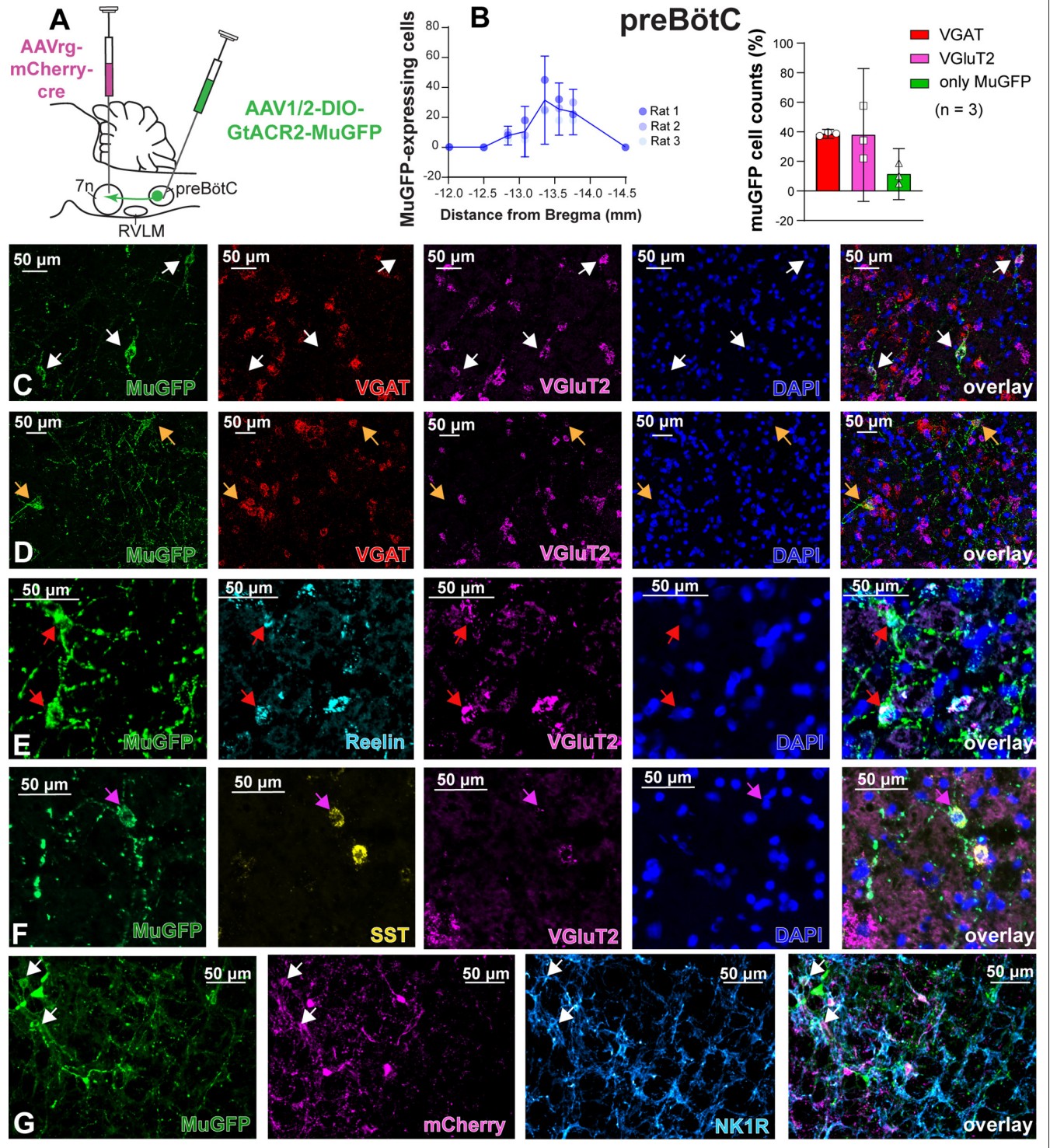

**Figure 5.** Excitatory and inhibitory pre-Bötzinger complex (preBötC) neurons are transduced by selective transduction of preBötC →7n neurons. (**A**) Schematic diagram showing the injection protocol for selective transduction of preBötC→7n neurons. (**B**) Quantification of the total number of MuGFP-expressing neurons, plotted as the distance from Bregma (mm). Note the slight trend toward a more caudal distribution. The histograms show the number of transduced cells that co-expressed mRNA for *VGAT* or *VGluT2*. The results are presented as mean ± 95% CI. (**C–F**) In situ hybridization showing the co-expression of MuGFP (green), with mRNA for VGAT (red), VGluT2 (magenta), reelin (cyan), or somatostatin (SST) (yellow) in preBötC→7n neurons. Nuclei are labeled in blue (DAPI). The yellow arrows highlight *VGAT* neurons, and the white arrows highlight *VGluT2* neurons, the red arrows highlight *reelin* neurons, and the pink arrows highlight *SST* neurons. (**G**) Immunohistochemistry showing that some neurons expressing MuGFP (green) and mCherry (magenta) also express neurokinin-1 receptor (NK1R) (light blue). Co-localization is indicated by white arrows.

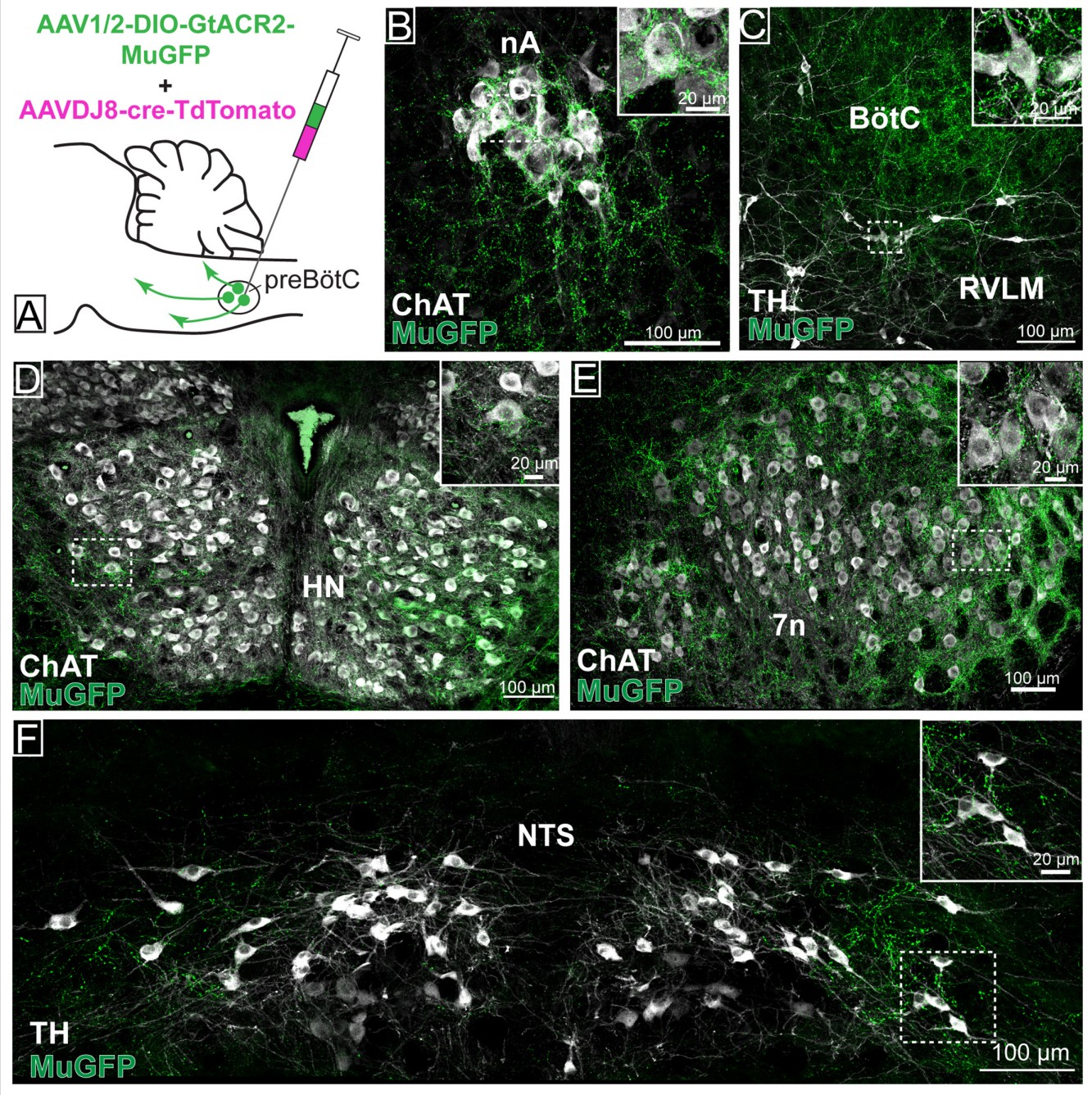

**Figure 6.** Distribution of GtACR2-MuGFP-expressing axons in multiple brainstem nuclei following the non-selective transduction of pre-Bötzinger complex (preBötC) neurons. (**A**) Schematic diagram showing the protocol for non-selective transduction of preBötC neurons. Confocal microscopy images demonstrate MuGFP expression in axon in the (**B**) nucleus ambiguus. (**C**) Rostral ventrolateral medulla and Bötzinger complex, (**D**) hypoglossal nucleus, (**E**) facial nucleus, and (**F**) nucleus of the solitary tract. Higher magnification images of the hashed-boxed regions are shown in the upper right corner of the lower magnification image. ChAT: choline acetyltransferase; TH: tyrosine hydroxylase.

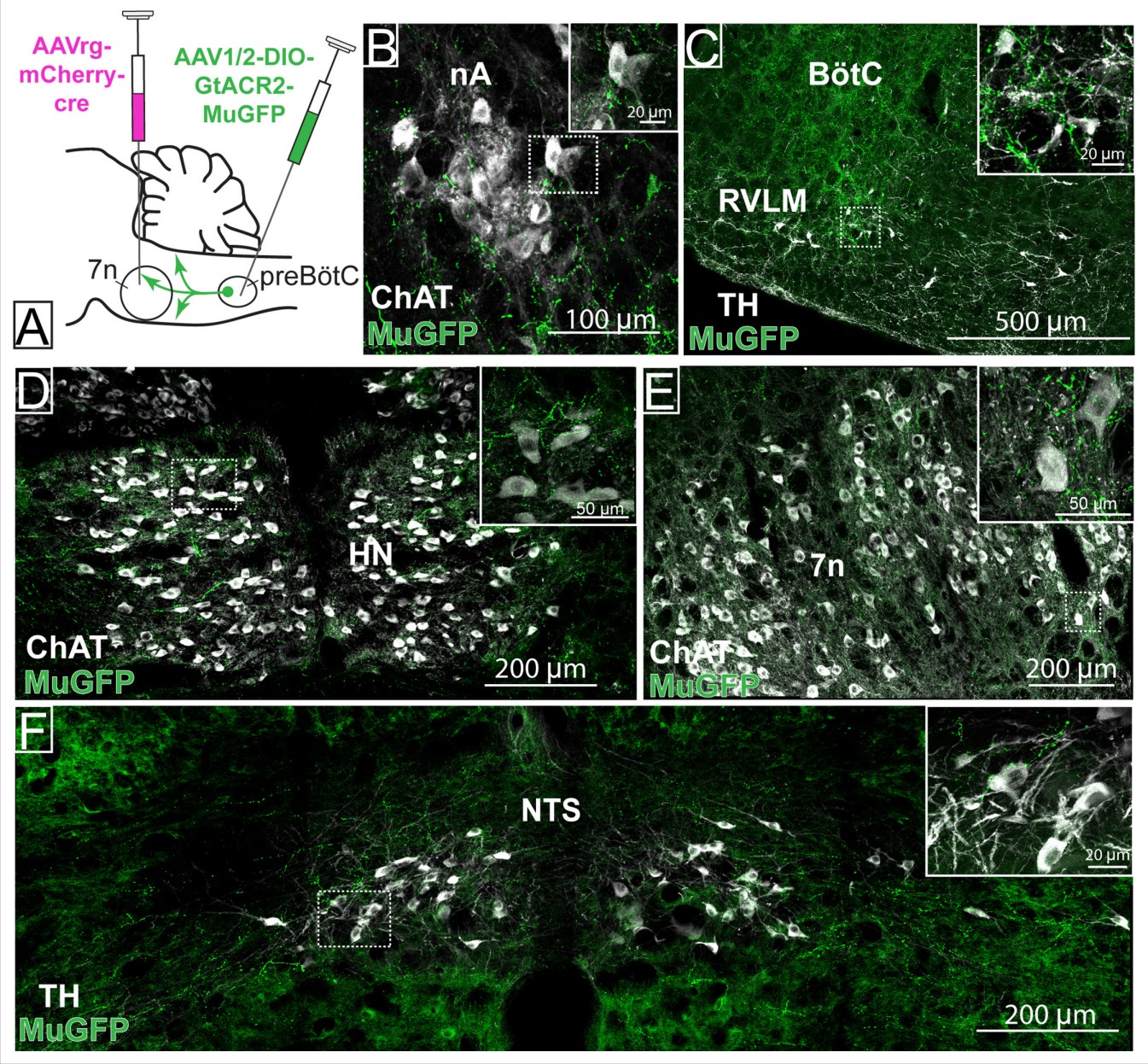

**Figure 7.** Distribution of selective pre-Bötzinger complex (preBötC)→7n GtACR2-MuGFP-expressing axonal projections. (**A**) Schematic diagram showing the injection protocol for selective transduction of preBötC→7n neurons. Confocal microscopy images highlighting the expression of MuGFP in axons in the (**B**) nucleus ambiguus; (**C**) rostral ventrolateral medulla and Bötzinger complex; (**D**) hypoglossal nucleus; (**E**) facial nucleus; and (**F**) nucleus of the solitary tract. Higher magnification images of the hashed-boxed regions are shown in the upper right corner of the lower magnification image. ChAT: choline acetyltransferase; TH: tyrosine hydroxylase.

activity (rat 1: 9.6 mV vs. baseline: 3.3 mV; rat 2: 17.6 mV vs. baseline: 8 mV) was observed in the other two rats tested in this experiment (*Figure 8—figure supplement 3B*). No changes were observed in HR (*Figure 8Q and R*).

## Conscious phase

Non-selective photoinhibition of preBötC neurons induced apnea and did not affect HR (*Figure 9A–I*). The amplitude of MP activity did not change, but became tonic (*Figure 9B and J*; *Figure 8—figure supplement 2C*). The overall activity of MP increased in the initial period of selective photoinhibition of preBötC→7n neurons in all animals tested in this experiment (*Figure 9L and T*), even in those

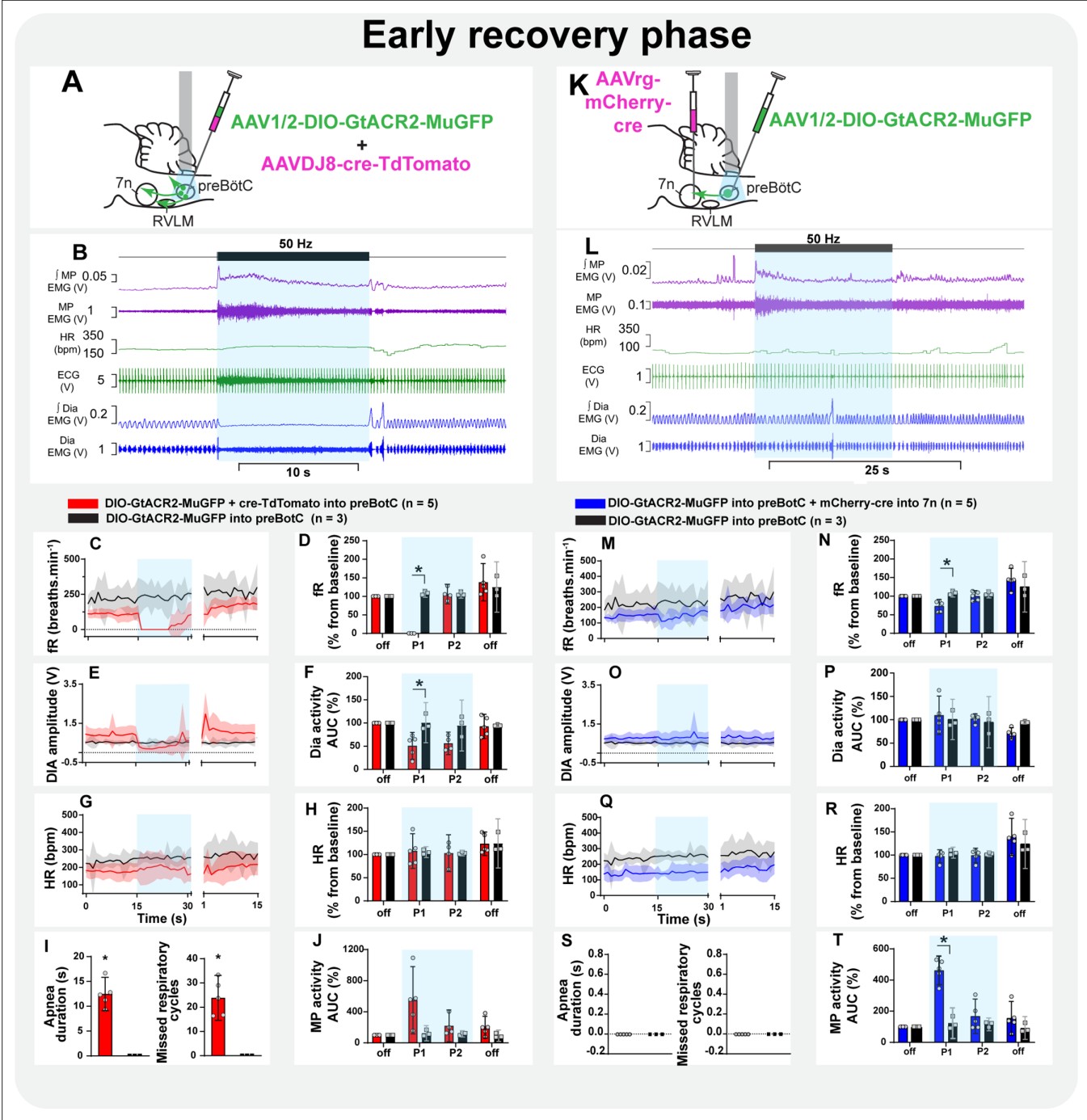

**Figure 8.** Effect of selective inhibition of pre-Bötzinger complex (preBötC)→7n neurons on mystacial pad (MP) activity is state-dependent. Schematic diagrams showing the injection protocols for non-selective transduction of preBötC neurons (**A**) and selective transduction of preBötC→7n neurons (**K**). Representative trace showing the integrated (∫) and raw MP EMG, heart rate (HR), electrocardiograms (ECG), and ∫ and raw diaphragm (Dia) EMG in the initial phase of reversal of ketamine/medetomidine anesthesia in rats with non-selective (**B**) and selective (**L**) preBötC neuron transduction. The period of photoinhibition is highlighted with the blue box. Group data showing mean (solid line) and 95% confidence intervals for (**C and M**) respiratory frequency - fR, (**E and O**) Dia amplitude, (**G and Q**) and HR (bpm) before, during, and after photoinhibition in non-selective (red), selective GtACR2 expressing (blue), and control (black) rats. Histograms showing group data for the effect of photoinhibition of preBötC on respiratory and cardiovascular parameters (**D and N**) fR, (**F and P**) Dia amplitude, (**H and R**) HR, (**I and S**) apnea duration and number of missed respiratory cycles, and (**J and T**) MP activity.; P1 and P2 refer to the initial period of photoinhibition. Group data are presented as mean ± 95% CI; unpaired t-test or nonparametric Mann-Whitney test with multiple comparisons using the Bonferroni-Dunn method, *p<0.05. The photoinhibition of preBötC neurons is depicted by blue

*Figure 8 continued on next page*

*Figure 8 continued*

shading. The effects of non-selective and selective photoinhibition of preBötC in ketamine/medetomidine-anesthetized rats are shown in *Figure 8—figure supplement 1*.

The online version of this article includes the following source data and figure supplement(s) for figure 8:

**Source data 1.** Source data and statistics for *Figure 8*.

**Figure supplement 1.** Effect on non-selective and selective inhibition of pre-Bötzinger complex (preBötC) on mystacial pad (MP) activity under ketamine/medetomidine anesthesia.

**Figure supplement 1—source data 1.** Source data and statistics for *Figure 8—figure supplement 1*.

**Figure supplement 2.** Effects of non-selective inhibition of pre-Bötzinger complex (preBötC) on the mystacial pad (MP) and diaphragm activity in different state conditions.

**Figure supplement 3.** Effects of selective inhibition of pre-Bötzinger complex (preBötC)→7n neurons on the mystacial pad and diaphragm activity in different state conditions.

**Figure supplement 3—source data 1.** Source data and statistics for *Figure 8—figure supplement 3*.

rats where the respiratory-modulated rhythm of MP activity was absent in the baseline (three out of five rats - *Figure 8—figure supplement 3C*). An expiratory-related activity of the mystacial pad was observed in the other two rats. Curiously, photoinhibition of preBötC→7n increased the magnitude of the expiratory activity of MP in one rat, it decreased in the other one (*Figure 8—figure supplement 3C*). Diaphragm EMG was not affected, except for the induction of a very short apnea in two out of the five rats, but respiratory frequency decreased throughout the entire stimulation period (*Figure 9L–P and S*). Inhibition of preBötC→7n neurons did not affect HR (*Figure 9Q and R*).

## Discussion

We employed a combinatorial Cre-dependent approach to transduce a subgroup of preBötC neurons based on their axonal projections to the facial nucleus. The preBötC is the kernel for breathing (*Smith et al., 1991*), but also acts as a master oscillator controlling cardiovascular and orofacial activities (*Del Negro et al., 2018*; *Deschênes et al., 2016*; *Huff et al., 2022*; *Takatoh et al., 2022*), and our study aimed to determine whether these functions involved independent subgroups of preBötC neurons. We validated our approach by co-injections of a Cre virus and the Cre-dependent anion channel virus - non-selective preBötC transduction. In agreement with previous results (*Menuet et al., 2020*), non-selective photoinhibition of preBötC neurons induced apnea, bradycardia, and biphasic effects on BP. We also observed loss of inspiratory modulation of mystacial pad activity, with increased tonic activity, except in rats anesthetized with ketamine/medetomidine where there was no ongoing mystacial pad activity. By contrast, selective photoinhibition of preBötC→7n neurons had minimal effect on breathing or cardiovascular activity, but altered mystacial pad activity with the effect altered by the type of anesthetic or state. The preBötC→7n neurons showed a restricted anatomical distribution within the preBötC but sent substantial collateral projections to several areas in the brainstem involved with the cardiorespiratory regulation. We conclude that these collaterals are functionally active and responsible for the small effects of photoinhibition of preBötC→7n neurons on BP, HR, and respiratory frequency. Both selective preBötC→7n neuron transduction and non-selective preBötC transduction resulted in transgene expression in both inhibitory and glutamatergic neurons.

The observation that photoinhibition of preBötC→7n neurons significantly affects mystacial pad activity, without affecting breathing, BP, or HR, clearly demonstrates that modulation of preBötC subgroups of neurons based on their axonal projections is a useful strategy. It allows an understanding of the anatomical distribution and neurochemical phenotype of subgroups of preBötC neurons. It also enables assessment of the physiological relevance of these subgroups without interference and potentially the development of secondary physiological changes that occur with profound interruption of breathing. The compartmentalization of the preBötC into segregated subgroups of neurons based on their connections impacts our comprehension of mechanisms that coordinate and integrate breathing with different motor and physiological behaviors. This is of fundamental importance, given that abnormal respiratory modulation of autonomic activity and orofacial behaviors have been associated with the development and progression of diseases (*El-Omar et al., 2001*; *Huff et al., 2022*; *Menuet et al., 2017*; *Simms et al., 2009*). For example, exaggerated respiratory-sympathetic

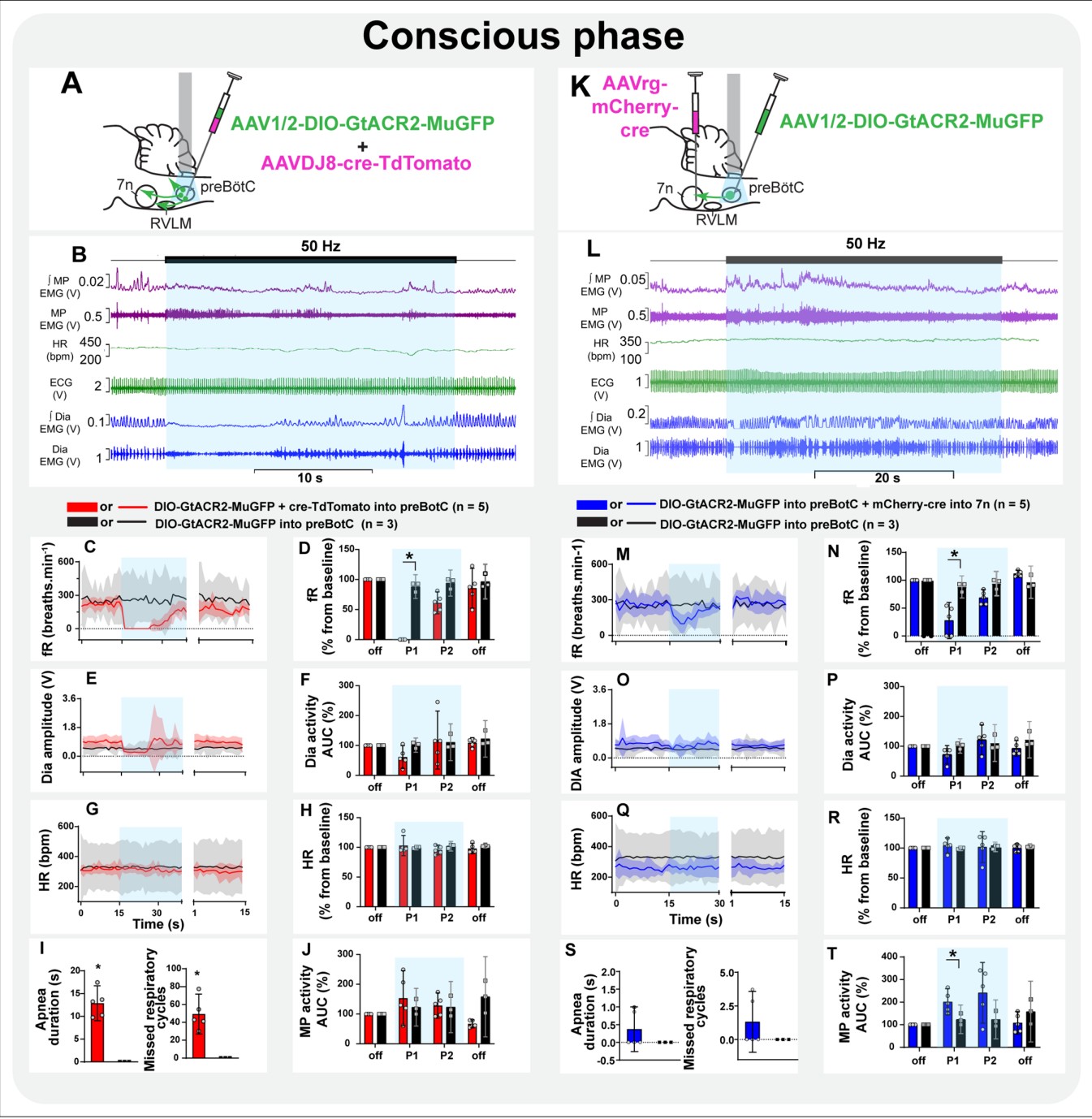

**Figure 9.** Inspiratory-related activity of mystacial pad is interrupted by the selective inhibition of pre-Bötzinger complex (preBötC) to 7n neurons in conscious rats. Schematic diagrams showing the injection protocols for non-selective transduction of preBötC neurons (**A**) and selective transduction of preBötC→7n neurons (**K**). Representative trace showing the integrated (∫) and raw mystacial pad (MP) EMG, heart rate (HR), electrocardiograms (ECG), and ∫ and raw diaphragm (Dia) EMG after recovery from ketamine/medetomidine anesthesia in rats with non-selective (**B**) and selective (**L**) preBötC neuron transduction. The period of photoinhibition is highlighted with the blue box. Group data showing mean (solid line) and 95% confidence intervals for (**C and M**) respiratory frequency - fR, (**E and O**) Dia amplitude, (**G and Q**) and HR (bpm) before, during, and after photoinhibition in non-selective (red), selective GtACR2 expressing (blue) and control (black) rats. Histograms showing group data for the effect of photoinhibition of preBötC on respiratory and cardiovascular parameters (**D and N**) fR, (**F and P**) Dia amplitude, (**H and R**) HR, (**I and S**) apnea duration and number of missed respiratory cycles, and (**J and T**) MP activity; P1 and P2 refer to the initial period of photoinhibition. Group data are presented as mean ± 95% CI; unpaired t-test or nonparametric Mann-Whitney test with multiple comparisons using the Bonferroni-Dunn method, *p<0.05. The photoinhibition of preBötC neurons is depicted by blue shading.

*Figure 9 continued on next page*

*Figure 9 continued*

The online version of this article includes the following source data for figure 9:

**Source data 1.** Source data and statistics for *Figure 9*.

modulation seems to be an important mechanism for the development and progression of hypertension (*Barnett et al., 2020*; *Menuet et al., 2017*; *Moraes et al., 2014*; *Simms et al., 2009*; *Tatasciore et al., 2007*). Reduced inspiratory modulation of HR is associated with the severity of diverse pathological conditions such as hypertension, heart failure, depression, and anxiety (*El-Omar et al., 2001*; *Masi et al., 2007*; *Thayer et al., 2012*). In addition, loss of the synchronization between breathing and orofacial and oropharyngeal behaviors, such as swallowing, has significant clinical implications, as it increases the risk of aspiration pneumonia associated with dysphagia, which can lead to death (*Beard et al., 1996*; *Heemskerk and Roos, 2012*; *Huff et al., 2022*).

Inhibition of preBötC→7n only transiently affects the mystacial pad activity despite the sustained laser light delivery. These results are similar to our previous study, which showed that non-selective photoinhibition of preBötC neurons transiently affected inspiration and sympathetic nerve activity, while the bradycardia, induced by vagal activation, was retained for the length of light delivery (*Menuet et al., 2020*). We do not know the mechanisms underlying this phenomenon and further investigation is needed. We can only speculate that intrinsic biophysical properties, such as maintenance of the chloride gradient, may differ between distinct neuronal populations, enabling some neurons to escape prolonged photoinhibition.

Nasofacial and orofacial activity are sensitive to anesthesia and animal state (*Deschênes et al., 2016*; *Deschênes et al., 2015*). Naris dilation and whisker motion are silent when animals are deeply anesthetized with ketamine/medetomidine but resume in synchrony with the inspiratory phase of the respiratory cycle, as rats recover from anesthesia (*Deschênes et al., 2015*). We also observed that mystacial pad activity is minimal and non-rhythmic under surgical anesthesia with ketamine/medetomidine, but resumes, with inspiratory modulation, as the animal recovers consciousness. In contrast, strong inspiratory-related mystacial pad activity occurs when surgical anesthesia is provided by urethane. In both cases, surgical anesthesia was defined by loss of the pedal withdrawal and corneal reflexes. The mechanisms by which deep ketamine and urethane anesthesia differentially affect respiratory-related motor activity are unknown. Under light ketamine anesthesia, GABAergic cells of vIRt displayed phase-locked firing patterns for protraction or retraction phases of whisking, while only retraction unit activity was recorded under light urethane anesthesia, suggesting that only inhibitory vIRt cells, whose activity is correlated with vibrissa retraction, are active in urethane-anesthetized rats (*Deschênes et al., 2016*).

Likewise, the effect of photoinhibition of preBötC→7n neurons on mystacial pad activity is both state- and anesthetic-dependent. Under urethane anesthesia, selective photoinhibition of preBötC→7n neurons silenced mystacial pad activity, while photoinhibition during different phases of recovery from ketamine/medetomidine anesthesia interrupted the inspiratory-related and increased the overall tonic mystacial pad activity. The reason for this substantial state-dependent effect of the preBötC→7n on mystacial pad activity remains unclear, but it has been reported in other systems. Injection of L-glutamate in the NTS induces a substantial pressor response in awake normotensive rats but a depressor response in the same rats anesthetized with urethane (*Machado and Bonagamba, 1992*). Photoinhibition of the C1-RVLM neurons in awake rats produces a very small depressor response (*Wenker et al., 2017*) when compared to that observed in anesthetized animals (*Marina et al., 2011*). Likewise, electrical stimulation of the central amygdala induces a significant pressor response in conscious rats but a depressor response under anesthesia (*Chiou et al., 2009*). Changes in the activity of several fast ionotropic transmitter systems, including glycine, GABA, acetylcholine, and glutamate, have been reported in urethane anesthesia (*Accorsi-Mendonça et al., 2007*; *Hara and Harris, 2002*). While speculative, we hypothesize that under urethane, the glutamatergic preBötC→7n neurons play a major role in regulating the respiration-related mystacial pad activity, while the inhibitory neurons are more active in the conscious state.

Under urethane anesthesia, and during the early recovery phase following ketamine/medetomidine anesthesia, inspiratory-related mystacial pad activity was clearly evident. In the conscious state, heterogeneous respiratory-related mystacial pad activity was observed. The electrodes we used to record mystacial pad activity were inserted near the rostral tip of the snout, which is known to be

the site of origin of the *nasolabialis profundus*. However, due to the large number of distinct muscles that compose the mystacial pad (*Haidarliu et al., 2010*), the difficulty of dissecting the rodent snout (*Haidarliu et al., 2012*), the substantial size of the electrode's suture pads, and the fact that the *nasolabialis profundus* electromyogenic activity was not recorded prior to the electrode implantation, it is possible that the mystacial pad activity we recorded reflects the activity of multiple inspiratory- and expiratory-related muscles. Nonetheless, given that *nasolabialis profundus* is active during basal respiration, while extrinsic muscles are silent, and preferentially contract during sniffing and whisking (*de Britto et al., 2020*; *Deschênes et al., 2015*), we conclude that the inspiratory component of the mystacial pad activity predominantly reflects the *nasolabialis profundus* muscle contraction during urethane anesthesia and the early recovery phase following ketamine/medetomidine. The absence of respiratory-related activity of the mystacial pad in some conscious rats was unexpected, but could be the result of sniffing and/or whisking behavior, as indicated by the high respiratory frequency.

Non-selective inhibition of preBötC induced apnea and immediate interruption of the inspiratory-related mystacial pad activity, which increased in amplitude and became tonic. The inspiratory-related mystacial pad activity recovered when breathing resumed. This coordinated activity between breathing and orofacial/nasofacial activity has been described before (*Deschênes et al., 2015*). It was demonstrated that both naris dilation and vibrissae retraction are abolished during apnea induced by the application of ammonia to the snout, and their activity synchronously recovers with the resumption of breathing. Nonetheless, the neural mechanisms that underly these responses are still unclear. Non-selective photoinhibition of preBötC neurons may lead to inhibition of preBötC→7n, and at the same time, to inhibition of preBötC inhibitory neurons that project to parvalbumin-expressing inhibitory neurons of the vIRt neurons (vIRt$_{PV}$). The vIRt$_{PV}$ was recently identified as the whisking oscillator (*Takatoh et al., 2022*), and direct projections from preBötC have been shown to reset the vIRt$_{PV}$ activity (*Deschênes et al., 2016*; *Golomb et al., 2022*; *Takatoh et al., 2022*). The vIRt$_{PV}$ neurons also receive local presynaptic inputs from non-PV-glutamatergic neurons, which are not required for whisking, and from non-PV-GABAergic neurons, whose role in whisking is unclear (*Takatoh et al., 2022*). The post-mortem analysis showed that the non-selective approach transduced some vIRt$_{PV}$ and non-PV-vIRt neurons, while the selective transduction only sparsely transduced the non-PV-vIRt neurons. Based on this observation, it is likely that the vIRt$_{PV}$ activity could have been directly affected by the laser delivery, producing desynchronization of facial motoneurons activity and suppression of the rhythmic whisking (*Takatoh et al., 2022*).

## Photoinhibition of preBötC neurons modulated expiratory abdominal muscle activity

Beyond affecting inspiratory and MP activity, non-selective inhibition of preBötC also increased and produced tonic activity of the abdominal muscle that lasted for the apnea duration. The active expiratory activity of the abdominal muscles is strictly controlled by quiescent and synaptically inhibited late-expiratory neurons located in the lateral parafacial nucleus (*Magalhães et al., 2021*; *Pagliardini et al., 2011*). Our results support the evidence that preBötC is a potential source of inhibitory input to lateral parafacial. This interaction could be essential for the generation of active expiration in situations with increased respiratory demand, such as hypercapnia/acidosis (*Del Negro et al., 2018*). Interestingly, in addition to regulating expiratory activity, it has been shown that the lateral parafacial nucleus may also play a role in coordinating nasofacial and orofacial behavior during high chemical drive via direct projections to 7n (*de Britto et al., 2020*). Thus, indirect projections from preBötC to 7n via lateral parafacial nucleus may also be involved with the responses on the mystacial pad activity induced by the non-selective inhibition of preBötC.

## Facial projecting preBötC neurons have functionally relevant collateral projections

While there was a clear difference in the effect of selective preBötC→7n photoinhibition on breathing, compared to non-selective preBötC photoinhibition, selective photoinhibition did have small effects on breathing, particularly under ketamine/medetomidine anesthesia and its recovery phases. With the selective approach, we observed a much smaller population of transduced neurons with a more restricted anatomical location. We did not see any transgene expression in the parvalbumin-expressing neurons of rVRG (*Alheid et al., 2002*; *Wu et al., 2017*). Both selective and non-selective approaches

resulted in the MuGFP expression in a similar proportion of inhibitory and glutamatergic neurons. Interestingly, SST, reelin, and NK1R neurons were transduced. These are known as biomarkers for respiratory preBötC neurons. These results raised the question of whether preBötC→7n neurons are part of preBötC respiratory network or if they are a distinct neuronal group. While our data are not sufficient to adequately address this question, our working hypothesis is that the facial nucleus projecting preBötC neurons are transmitting respiratory rhythm, rather than being responsible for generating it. We conclude that the small effect of photoinhibiting these neurons on inspiratory activity is due to collaterals. However, we cannot rule out the possibility that the neurons projecting to the facial nucleus are a small subgroup of the rhythm-generating neurons with collaterals to the facial nucleus. In such a case, the effects on mystacial pad activity could be indirectly caused by the slowing of breathing. This hypothesis could be addressed by inhibiting the axons of preBötC→7n neurons within the facial nucleus. Although we have contemplated directing the optical fibers to the facial nucleus to evaluate this, we are cautious about using GtACR2 to inhibit axonal terminals, given the evidence of increased synaptic activity and neurotransmitter release in response to light-induced activation of GtACR2 at the axon terminal (*Messier et al., 2018*).

We observed that the transduced axons of preBötC→7n neurons projected widely into multiple brainstem nuclei, including RVLM, nA, NTS, BötC, rVRG, and vIRt. Collateral projections from excitatory and inhibitory preBötC neurons to premotor and motoneurons have been described before and appear to be essential for the coordination of inspiratory and expiratory activity (*Koizumi et al., 2013*). It was suggested that divergent axonal projections from the commissural excitatory preBötC into the HN and rVRG may be essential for initiating the inspiratory activity. On the other hand, feedforward inhibition via collateral projections from inhibitory preBötC into these same regions may contribute to the dynamic shaping of the respiratory pattern (*Koizumi et al., 2013*). While the functional role of many of these collateral projections has not been established yet, we hypothesize that the small effect of selective photoinhibition reflects their ongoing, functionally relevant activity. It is possible that the strict synchronization between premotor and motor neurons is not confined to respiratory motor activity, but is crucial for the coordination of respiratory, cardiovascular, and orofacial/nasofacial activity necessary for the execution of complex behaviors, such as exercise, response to stress, or pain.

## Ideas and speculation

The preBötC, initially defined as the kernel for generating inspiratory rhythm, appears to act as a master oscillator regulating other physiological functions, such as orofacial behaviors and autonomic nervous activity. Our study suggests that groups of these neurons play principal roles in these specific functions. However, the widespread collateralization we observed along with effects of photoinhibition of the preBötC→7n on disparate motor activities raises the possibility that preBötC neurons may coordinate multiple outputs. Whether this is a function of most neurons projecting to more than one target, or a small number of neurons projecting to many targets, remains to be determined.

As has been described previously using transgenic mice (*Yang and Feldman, 2018*), we also observed that different subpopulations of preBötC→7n express either markers of an excitatory phenotype or an inhibitory phenotype. The idea that one output nucleus provides both excitatory and inhibitory projection to a target is intriguing. Our methods could not determine whether these different neurochemical groups might project to the same target neuron, or whether the pathways are parallel and separate. It is possible that the different groups are active under different states and enable altered coupling to the respiratory cycle. For example, we speculate that under urethane anesthesia, when mystacial pad EMG is active, glutamatergic preBötC→7n neurons play a major role in regulating the respiration-related mystacial pad activity and their photoinhibition resulted in a decrease in activity. In contrast, inhibitory preBötC→7n neurons would be more active in the conscious state, and their inhibition resulted in an increase in activity.

Exaggerated inspiratory modulation of sympathetic activity is associated with the development of hypertension (*Menuet et al., 2017*; *Simms et al., 2009*), which can increase BP variability. Interestingly, independent of the BP level, BP variability seems to be an important contributor to organ damage, such as renal dysfunction and left ventricular hypertrophy, cardiovascular disease, and poor clinical outcomes (*Messerli et al., 2019*). The central network underlying this autonomic dysfunction remains to be elucidated. Our previous studies suggest that respiratory input from preBötC to the catecholaminergic C1 neurons of RVLM may be involved with the increased inspiratory-sympathetic

modulation in hypertensive rats (*Menuet et al., 2017*). In this way, methodologies that only affect inputs from preBötC to C1 neurons, such as the approach used in the present study, could shine the light on the mechanisms involved in the development of hypertension and possibly contribute to the development of targeted therapeutics used to prevent hypertension development.

## Conclusion

We tested the hypothesis that the preBötC might consist of separate subpopulations of neurons that project to specific nuclei to coordinate respiratory rhythmicity with different physiological behaviors, such as nasofacial activity. We showed that even when selecting just neurons projecting to a specific target, both excitatory and inhibitory neurons were transduced. Selective photoinhibition of these neurons enabled observation of the effect on nasofacial motor activity in the absence of substantial changes in respiratory, or other autonomic, activities. However, small effects on these other functions, such as diaphragm muscle activity, remain. We observed collateral axonal projections of preBötC→7n neurons to several brainstem nuclei, including the rVRG, and conclude that these are functional, active projections. This unmasks the possibility that these neurons may play a role in the complex synchronization between respiratory, cardiovascular, orofacial, and potentially other, physiological functions.

# Materials and methods

**Key resources table**

| Reagent type (species) or resource | Designation | Source or reference | Identifiers | Additional information |
|---|---|---|---|---|
| Strain, strain background (male Sprague-Dawley rats) | WT Sprague-Dawley rats | Biomedical Science Animal Facility of the University of Melbourne | | |
| Recombinant DNA reagent | pAAV-hSyn-DIO-(hCAR)off-(ChETA-mRuby2)on-W3SL | Addgene, gift from Prof. Adam Kepecs | Addgene plasmid #111391 | |
| Strain, strain background (AVV) | AAV1/2-CAG-DIO-(hCAR)off-(GtACR2Kv2.1)on-W3SL | In-house cloning and virus production | | $2.04 \times 10^{11}$ GC/ml |
| Strain, strain background (AVV) | AAVDJ8-CBA-Cre-TdTomato-WPRE | In-house cloning and virus production | | $1.14 \times 10^{12}$ GC/ml |
| Strain, strain background (AVV) | AAVrg-EF1α-mCherry-IRES-cre | Addgene | Addgene # 55632 | $8 \times 10^{12}$ GC/ml |
| Strain, strain background (AVV) | AAVrg-CAG-GFP | In-house cloning and virus production | | $1.29 \times 10^{13}$ GC/ml |
| Antibody | Chicken polyclonal anti-GFP | Aves Labs Inc | Cat#: GFP-1010 | 1:5000 |
| Antibody | Mouse monoclonal anti-TH | Merck-Millipore | Cat#: MAB318 | 1:5000 |
| Antibody | Rabbit polyclonal anti-parvalbumin | Abcam | Cat#: ab11427 | 1:5000 |
| Antibody | Goat polyclonal anti-ChAT | Chemicon-Merck | Cat#: AB144P | 1:1000 |
| Antibody | Rabbit polyclonal anti-dsRed | Takara Bio, Clontech | Cat#: 632496 | 1:5000 |
| Antibody | Goat polyclonal anti-mCherry | Sicgen, Cantanhede | Cat#: AB0040-200 | 1:5000 |
| Antibody | Alexa488-conjugated donkey polyclonal anti-chicken | Jackson ImmunoResearch Laboratories, Inc | Cat#: 703545155 | 1:500 |
| Antibody | Cy3-conjugated donkey polyclonal anti-rabbit | Jackson ImmunoResearch Laboratories, Inc | Cat#: 711165152 | 1:500 |
| Antibody | Cy3-conjugated donkey polyclonal anti-goat | Jackson ImmunoResearch Laboratories, Inc | Cat#: 705165003 | 1:500 |
| Antibody | Cy5-conjugated donkey polyclonal anti-mouse | Jackson ImmunoResearch Laboratories, Inc | Cat#: 715175151 | 1:500 |
| Antibody | Cy5-conjugated donkey polyclonal anti-goat | Jackson ImmunoResearch Laboratories, Inc | Cat#: 705175147 | 1:500 |

*Continued on next page*

*Continued*

| Reagent type (species) or resource | Designation | Source or reference | Identifiers | Additional information |
|---|---|---|---|---|
| Antibody | Biotin-SP-conjugated donkey polyclonal anti-goat | Jackson ImmunoResearch Laboratories, Inc | Cat#: 705065147 | 1:500 |
| Antibody | Rabbit polyclonal anti-NK1R | Sigma-Aldrich | Cat # S8305 | 1:5000 |
| Antibody | Streptavidin-Marina blue | Invitrogen-Thermo Fisher Scientific | Cat#: S11221 | 1:200 |
| Commercial assay or kit | RNAscope multiplex fluorescence v1 kit | ACD | Cat #320851 | |
| Commercial assay or kit | Protease IV | ACD | Cat #322336 | |
| Commercial assay or kit | VGAT (SLC32A1) | ACD | Cat #ADV424541-C2 | |
| Commercial assay or kit | VGluT2 (Slc17a6) | ACD | Cat #317011-C3 | |
| Commercial assay or kit | SST | ACD | Cat#ADV412181-C3 | |
| Commercial assay or kit | Reelin | ACD | Cat ADV1048921- C1 | |
| Commercial assay or kit | C1 diluent | ACD | Cat #ADV300041 | |
| Chemical compound, drug | Meloxicam | Lyppard | | 1 mg/kg, s.c. |
| Chemical compound, drug | Isoflurane | Rhodia Australia | | 5% induction, 2–2.5% maintenance |
| Chemical compound, drug | Ketamine | Lyppard | | 75 mg/kg, i.p. |
| Chemical compound, drug | Medetomidine | Pfizer Animal Health | | 0.5 mg/kg, i.p. |
| Chemical compound, drug | Atipamazole | Pfizer Animal Health | | 1 mg/kg, i.p |
| Chemical compound, drug | Urethane | Sigma-Aldrich | | 1.4 g/kg, i.v. |
| Chemical compound, drug | Buprenorphine | Schering-Plough | | 0.025 mg/kg, s.c. |
| Software, algorithm | Spike2 | Cambridge Electrical Design | | |
| Software, algorithm | Zen Blue | Carl-Zeiss | | |
| Software, algorithm | Zen Black | Carl-Zeiss | | |
| Software, algorithm | ImageJ | NIH | | |
| Software, algorithm | SigmaPlot v11 | Systat Software Inc | | |
| Software, algorithm | Prims v9.0 | GraphPad | | |

## Animal experiments

Experiments were conducted in accordance with the National Health and Medical Research Council of Australia's 'Guidelines to promote the well-being of animals used for scientific purposes: The assessment and alleviation of pain and distress in research animals (2008)' and 'Australian code for the care and use of animals for scientific purposes' and were approved by the University of Melbourne Animal Research Ethics and Biosafety Committees (ethics ID#21396). All experiments were performed on male SD rats, initially weighing 60–80 g. The rats were housed in standard cages in groups of up to 4, had ad libitum access to standard chow and tap water, and were maintained under a 12:12 hr light-dark cycle in a 21°C temperature-regulated room at the University of Melbourne Biological Resources Facility.

## Plasmid design and generation of AAV-DIO-GtACR2-MuGFP

C-terminal fusion of a 65-amino acid trafficking motif from the voltage-gated potassium channel Kv2.1 (*Lim et al., 2000*) has been shown to enrich the expression of the *Gt*ACR2 in the somatic membrane of mouse cortical neurons (*Mahn et al., 2018*; *Messier et al., 2018*). We employed an analogous design in the present study to develop a *Gt*ACR2$^{Kv2.1}$ fusion construct, with a monomeric, ultra-stable green fluorescent protein (MuGFP) tag (*Scott et al., 2018*) linked with four alanine residues to the *Gt*ACR2 C-terminus. To achieve Cre-dependent expression, *Gt*ACR2$^{Kv2.1}$-MuGFP was incorporated

into the pAAV-hSyn-DIO-(hCAR)off-(ChETA-mRuby2)on-W3SL plasmid (Addgene plasmid #111391; *Li et al., 2018*); a gift from Prof. Adam Kepecs (Cold Spring Harbor Laboratory, NY, USA), using NheI and PacI restriction sites to replace the ChETA-mRuby2 coding sequence. The hSyn promoter was subsequently excised, and a CAG promoter sequence was ligated in its place via XbaI restriction sites. Following Cre-mediated recombination, the orientation of $GtACR2^{Kv2.1}$-MuGFP coding sequence is reversed relative to the promoter, enabling the expression of $GtACR2^{Kv2.1}$-MuGFP.

Co-transfection of pAAV-CAG-DIO-(hCAR)$_{off}$-($GtACR2^{Kv2.1}$-MuGFP)$_{on}$-W3SL with pDP1 and pDPII plasmids (*Grimm et al., 2003*) into AAV293 cells (Agilent Technologies, CA, USA) preceded harvesting and iodixanol gradient purification (as described by *Zolotukhin et al., 1999*, and *Ganella et al., 2013*, of AAV1/2-CAG-DIO-(hCAR)$_{off}$-($GtACR2^{Kv2.1}$-MuGFP)$_{on}$-W3SL vector - also referred to here as AAV-DIO-GtACR2-MuGFP. Titration of purified AAV vector was performed using quantitative polymerase chain reaction as described by *Ma et al., 2017*), using forward (5'-CATTCTCGGACACAAACTGG AGTACAAC) and reverse (5'- GTCTGCTAGTTGAACGGAACCATCTTC) primers targeting the MuGFP coding sequence, rather than WPRE (which is replaced here by W3SL). Primers were synthesized by Bioneer Pacific (VIC, Australia). GtACR2-MuGFP, Kv2.1, and CAG sequences were synthesized by GenScript (NJ, USA). Restriction enzymes and T4 DNA ligase were sourced from New England Biolabs (VIC, Australia) and Promega (NSW, Australia), respectively, and used according to the manufacturer's recommendations.

## Other viruses

Non-selective expression of GtACR2$^{Kv2.1}$-MuGFP in the preBötC was achieved by the injection of a mixture containing AAV-DIO-GtACR2-MuGFP (2.04×10$^{11}$ GC/ml) and AAVDJ8-CBA-Cre-TdTomato-WPRE (1.14×10$^{12}$ GC/ml) into preBötC. Selective expression of GtACR2-MuGFP-Kv2.1 in preBötC neurons that project to the 7n was obtained by injections of AAV-DIO-GtACR2-MuGFP into the preBötC and the retrograde pseudotyped AAVrg-EF1α-mCherry-IRES-Cre (8×10$^{12}$ GC/ml; Addgene #55632-AVVrg) into the 7n.

## Microinjection into the brainstem

Animals were anesthetized with an intraperitoneal injection of ketamine (75 mg/kg, i.p., Lyppard, Dingley, Australia) and medetomidine (0.5 mg/kg, i.p., Pfizer Animal Health, West Ryde, Australia). Eye moisture was maintained by the application of a hydrating gel (POLY VISC Eye Ointment, Alcon). The surgical field was shaved and disinfected with 80% ethanol and chlorhexidine. Surgery was initiated once a deep surgical level of anesthesia was obtained, as evidenced by loss of the pedal withdrawal and corneal reflexes. Throughout the protocol, body temperature was maintained at 37.5°C with a heating pad (TC-1000 Temperature controller, CWE Inc) that was covered with an autoclaved non-absorbent pad. Rats were then placed in a stereotaxic frame with the nose ventro-flexed (incisor bar –15 mm; RWD Life Science). Extracellular recordings of multiunit activity were used to functionally map the preBötC, as previously described (*Menuet et al., 2020*). For all injections, the pipette was 20° angled in the caudal-rostral axis, the tip pointing forward. For injections into preBötC, the pipette was descended into the brainstem 1.5 mm lateral to the midline and the injections were made at the most rostral point at which vigorous inspiratory-locked activity was isolated (typically 0.2±0.1 mm rostral, 1.6±0.4 mm ventral to the *calamus scriptorius*). The injections into preBötC were randomized and were made bilaterally, and a picospritzer (World Precision Instruments, Sarasota, FL, USA) was used to microinject the virus (50 nl over 5 min on each side).

Antidromic field potentials, elicited by stimulating the mandibular branch of the facial nerve, were used to map the dorsal, caudal, ventral, and lateral edges of the facial nucleus, allowing precise targeting of the RVLM and the facial nucleus, as described before (*Menuet et al., 2017*). Briefly, an electrode was positioned so that this gently touched the mandibular branch of the facial nerve. An electrical current (2.0±0.5 mA) was applied in order to stimulate the nerve. A picospritzer (World Precision Instruments, Sarasota, FL, USA) was used to microinject AAVrg (40 nl per injection site over 5 min) into the RVLM immediately caudal to the facial nucleus. Injections were made bilaterally, 1.6 mm lateral to the midline at the caudal edge of the facial nucleus and 3.5±0.4 mm ventral to the *calamus scriptorius*. These injections were made in two rostrocaudal levels separated by 300 μm; the most rostral being at the caudal edge of the facial nucleus. A picospritzer was also used to microinject AAVrg (40 nl per injection site over 5 min) into the facial nucleus. The injections were made

bilaterally and targeted the lateral and dorsal-lateral borders of the facial nucleus (1.9 lateral to the midline, +1.4 and +1.1 rostral, and –2.9±0.2 mm ventral to the *calamus scriptorius*). At the end of the surgery, the neck muscles and the cheek skin were sutured with sterile absorbable polyglycolic acid 4/0 (Surgicryl) sutures using simple interrupted stitches. Subcutaneous injection of non-steroidal analgesic (meloxicam 1 mg/kg, s.c., Metacam, Lyppard) and 1 ml of warmed Hartmann's solution for fluid replacement was performed just before the intraperitoneal administration of atipamezole (1 mg/kg, i.m., Antisedan, Pfizer Animal Health, West Ryde, Australia) to reverse the effect of anesthesia. Post-operative analgesia with meloxicam and buprenorphine (0.025 mg/kg, s.c., Schering-Plough, USA) was maintained during the next 48 hr, and rats were monitored for any signs of surgical complications and weighed every day for 14 days.

## Optogenetic experiments in urethane-anesthetized rats

Following the 3–4 weeks of recovery from stereotaxic surgery, anesthesia was induced by inhalation of isoflurane (Rhodia Australia Pty. Ltd.) in an enclosed chamber. Once anesthetized, rats were transferred to the surgical bench, where anesthesia was maintained with 2–2.5% isoflurane, delivered in oxygen using a SomnoSuite low-flow anesthesia delivery system (Kent Scientific). Body temperature was maintained at 37.5°C with a TC-1000 heat pad (CWE Inc). A pulse oximeter probe was placed on a paw, and the body temperature was monitored using a rectal probe attached to the SomnoSuite anesthesia equipment. For dEMG recordings, a lateral abdominal incision was made, and two nylon-insulated stainless-steel wire electrodes (0.25 mm insulated diameter) ending with suture pads (0.7 × 1.0 × 3.2 mm$^3$, Plastics One, VA, USA) placed in the costal diaphragm, 3–4 mm apart. For abdominal EMG recordings, a lateral transverse incision was made in the abdomen region, and two silver wire electrodes (0.635 mm insulated diameter, A-M Systems) were implanted in the oblique abdominal muscles, 4–5 mm apart. For mystacial pad EMG, an incision was made along the midline of the nose, and two silver wires (0.38 mm insulated diameter, A-M Systems) were implanted into the mystacial pad near the rostral area of the snout - approximately between rows C and D and columns 5–7, 3–4 mm apart. The femoral artery and vein of the left leg were cannulated for measurement of arterial pressure and drug administration, respectively (PE10 tubing [ID 0.28× OD 0.61 mm] connected to PE50 [0.17×1.45 mm]). Isoflurane anesthesia was gradually replaced by 1.2 g/kg intravenous urethane, following which rats were tracheotomized, and oxygen (100%) was directed over the tracheotomy cannula. The arterial catheter was connected to a Statham Gould (P23 Db) pressure transducer, and the dEMG, AbdEMG and mystacial pad EMG electrode wires were coupled to an amplifier (10 Hz to 5 kHz band-pass filter, 5 kHz sampling rate, Model 1700 Differential AC amplifier, A-M systems, Sequim, WA, USA). Arterial pressure and electromyography were recorded using Spike2 software (Cambridge Electronic Design). Rats were transferred to a stereotaxic frame (incisor bar +3 mm; RWD Life Science). To perform the optogenetic experiments, the optical fibers (200 μm diameter, 0.22 NA, Ø1.25 mm, RWD Life Science) were secured to a manipulator on the stereotaxic frame and lowered through the dorsal surface of the skull through burr holes made bilaterally in the occipital bone centered approximately 1 mm rostral to the occipital suture, 1.8±0.2 mm lateral from the midline. Output intensity from the optical fibers was determined using a PM100D Meter (Thorlabs, NJ, USA). The strongest physiological responses to *Gt*ACR2 stimulation (50 Hz, 5 ms pulse, 15–20 mW) were found at 7.5±0.5 mm ventral to the surface of the skull.

## Optogenetic experiments in conscious rats

Following the 3–4 weeks of recovery from the virus injection, animals were anesthetized with an intraperitoneal injection of ketamine (75 mg/kg) and medetomidine (0.5 mg/kg). Eye moisture was maintained by the application of a hydrating gel (POLY VISC Eye Ointment, Alcon). The surgical field was shaved and disinfected with 80% ethanol and chlorhexidine. Surgery was initiated once a deep surgical level of anesthesia was obtained, as evidenced by loss of the pedal withdrawal and corneal reflexes. Throughout the protocol, body temperature was maintained at 37.5°C with a heating pad (TC-1000 Temperature Controller, CWE Inc) covered with an autoclaved non-absorbent pad. Then, the rats underwent the implantation of a six-channel pedestal with a back-mount (Type 363 components, PlasticsOne, USA) that contains percutaneous electrodes (2× cardiac electrocardiograms [ECG], 2× dEMG and 2× mystacial pad EMG) and remained externalized for tethering. The electrodes have socket-pin connections at the pedestal and stainless-steel suture pads (~0.5 mm$^2$) for electrical recording

(PlasticsOne, USA). For implantation of the sterilized multi-lead electrode pedestal, a transverse incision was made on the dorsal surface, approximately 1 cm below the rat's shoulder blades, and a subcutaneous tunnel opened between the two incisions. The two ECG and dEMG electrodes were placed according to previous studies (**Butler et al., 2021**; **O'Callaghan et al., 2020**). The mystacial pad EMG electrodes were fixed with a non-absorbable suture into the mystacial pad near the rostral area of the snout - approximately between rows C and D and columns 5–7, 3–4 mm apart. The skin incision was closed with sterile absorbable polyglycolic acid 4/0 (Surgicryl) sutures and the multi-lead electrode pedestal was sutured with sterile non-absorbable surgical sutures (4–0 Supramid) into place between the scapulae. All electrodes were tunneled to the pedestal and connected to it. At this point, the multi-lead electrode pedestal was connected to an amplifier (10 Hz to 5 KHz band-pass filter, 5 kHz sampling rate, Model 1700 Differential AC amplifier, A-M systems, Sequim, WA, USA). The signal was recorded and integrated using Spike2 version 9.0 software (Cambridge Electronic Design).

Following the implantation of the electrodes, rats were transferred to a stereotaxic frame (incisor bar +3 mm; RWD Life Science). To perform the optogenetic experiments, the optical fibers (200 µm diameter, 0.22 NA, Ø1.25 mm, RWD Life Science) were secured to a manipulator on the stereotaxic frame and lowered through the dorsal surface of the skull through burr bilateral holes made in the occipital bone centered approximately 1 mm rostral to the occipital suture, 1.8±0.2 mm lateral from the midline. Output intensity from the optical fibers was determined using a PM100D Meter (Thorlabs, NJ, USA). Physiological responses to GtACR2 stimulation were investigated at 0.5 mm increments moving ventrally from the surface of the skull. The strongest physiological responses to *Gt*ACR2 stimulation (50 Hz, 5 ms pulse, 15–20 mW) were typically observed at 7.5±0.5 mm ventral, and it was where the optical fibers were fixed with dental cement (Pattern Resin LS, GC America INC, IL, USA).

The rat was transferred to a recording cage, and the effects of preBötC photoinhibition on dEMG, ECG, and mystacial pad EMG were evaluated under three different states: when the animals were still under ketamine/medetomidine anesthesia, during the very early stage of regaining consciousness, a few minutes after the anesthesia was reversed with atipamezole and 1–2 hr after the atipamezole injection when the animal had recovered from anesthesia, was completely awake and freely behaving.

## Immunohistochemistry

At the end of the urethane-anesthetized and conscious experiments, animals were perfused transcardially with 0.1 M phosphate-buffered saline (PBS) (1 ml/g of b.w.) followed by 4% formaldehyde in 0.1 M PBS using a peristaltic pump (Masterflex L/S Drive System, Cole-Parmer, Vernon Hills, IL, USA). The brains were removed, post-fixed in 4% PFA for 12 hr at 4°C, and then immersed in 20% sucrose at 4°C until processing. Brainstems were frozen at –20°C, and four series of coronal sections (40 µm thickness) were cryostat sectioned. Sections were placed directly into cryoprotectant in a 24-well plate and stored at −20°C until processing. Fluorescence immunohistochemistry was performed as previously described (**Bassi et al., 2022**; **Menuet et al., 2020**; **Ngo et al., 2020**). Primary antibodies used were chicken anti-GFP (1:5000, Aves Labs Inc, Davis, CA, USA, Cat#: GFP-1010), mouse anti-TH (1:5000, Merck-Millipore, Bayswater, VIC, Australia, Cat#: MAB318), rabbit anti-parvalbumin (1:5000, Abcam, Melbourne, VIC, Australia, Cat#: ab11427), rabbit anti-NK1R (1:5000, Sigma-Aldrich, Melbourne, VIC, Australia, Cat # S8305), goat anti-ChAT: (1:1000, Chemicon-Merck, Bayswater, Australia, Cat#: AB144P), rabbit anti-dsRed: (1:5000, Takara Bio, Clontech, Australia, Cat#: 632496), goat anti-mCherry (1:5000, Sicgen, Cantanhede, Portugal; Cat#: AB0040-200).

All secondary antibodies were purchased from Jackson ImmunoResearch Laboratories, Inc, West Grove, PA, USA, and were used at either 1:500. The following secondary antibodies were used: Alexa488-conjugated donkey anti-chicken (Cat#: 703545155), Cy3-conjugated donkey anti-rabbit (Cat#: 711165152), Cy3-conjugated donkey anti-goat (Cat#: 705165003), Cy5-conjugated donkey anti-mouse (Cat#: 715175151), and Cy5-conjugated donkey anti-goat (Cat#: 705175003), and biotin-SP-conjugated donkey anti-goat (Cat#: 705065147). Streptavidin-Marina blue (1:200, Cat#: S11221) was purchased from Invitrogen-Thermo Fisher Scientific, Scoresby, VIC, Australia.

## In situ hybridization - RNAscope

Brains from SD rats with the selective and non-selective expression of GtACR2-MuGFP were used to perform this analysis. Briefly, brains were freshly extracted (n=3 per group) and frozen in isopentane and dry ice, then stored at –80°C. Brainstems were cryostat sectioned (16 µm), collected, and

mounted onto Superfrost-Plus slides (Thermo Scientific). Frozen sections were fixed in 4% PFA for 16 min at 4°C, rinsed twice in DEPC-treated 1× PBS for 1 min each, then dehydrated in 50%, 70%, and 100% ethanol solutions for 5 min each. Dehydrated sections were then stored in 100% ethanol overnight at –20°C. The following day, slides were air-dried at room temperature (RT), and hydrophobic barriers were created using ImmEdge hydrophobic PAP pen (Vector Laboratories, Burlingame, CA, USA). Sections were incubated with Protease IV (ACD, CA, USA; Cat #322336) for 20 min at RT, rinsed twice with DEPC-treated 1× PBS for 1 min each before incubating with the probe mixture for 2 hr at 40°C. The probe mixture consisted of probes against VGAT (SLC32A1-ACD;Cat #ADV424541-C2), VGluT2 (Slc17a6, ACD; Cat #317011-C3), SST (Sst- ACD; Cat#ADV412181-C3), reelin (Reln-ACD; Cat ADV1048921- C1), and C1 diluent (ACD, Cat #ADV300041). Signal amplification was achieved in accordance with the RNAscope multiplex fluorescence v1 kit manufacturer's instructions (ACD, Cat #320851).

Immediately following the RNAScope protocol, the immunofluorescence protocol for MuGFP was performed as previously described (*Bassi et al., 2022*), nuclei stained with 4',6-Diamidine-2-phenylindole dihydrochloride (DAPI), and coverslipped using ProLong Gold Antifade Mountant (Invitrogen by Thermo Fisher Scientific).

## Imaging

Imaging was performed at the Biological and Optical Microscopy Platform, University of Melbourne as previously described (*Bassi et al., 2022*). Lower power images were taken using epifluorescence microscopy (Zeiss Axio Imager D1 microscope with a Zeiss AxioCam MR3 camera and EC Plan-NeoFuar 10×/0.3 air objective, Carl-Zeiss, North Ryde, NSW, Australia). High-resolution images were captured on Zeiss LSM880 confocal laser scanning microscope with Airyscan detectors (32 GaAsP PMT array) at 20× using a Plan-Apochromat 20×/0.8 M27 objective (Carl-Zeiss, North Ryde, NSW, Australia) and ZEN 2.3 (black edition) imaging software. Z-stack images were taken with 0.4 µm z-intervals, and at least 50% overlap between optical slices obtaining 40–50 images over 15–30 µm thickness. Tile scan was performed with 10% overlap between neighboring tiles to image the entire distribution of labeled axonal fibers in the section. High-resolution images were also captured with high magnification using a Plan-Apochromat 63×/1.4 Oil DIC M27 objective.

## Mapping of reporter expression

While GtACR2-MuGFP expression was examined across all animals, the maps of the distribution were generated from seven and five representative animals of non-selective and selective expression of GtACR2-MuGFP, respectively. The animal was included in the data analysis only whether muGPF expression was confined within preBötC. To build the maps of distribution, the images containing GtACR2-MuGFP expression were selected and transformed in binary images according to a previously determined protocol (*Bassi et al., 2022*). The traced bitmap image was overlaid on the corresponding coronal preBötC schematic diagram (*Palkovits, 1983*) using the original fluorescence image as a guide. The total number of MuGFP immunofluorescent cells was determined either manually or using a macro every 160 µm. The quantification of co-localization among GtACR2-MuGFP, VGAT, and VGlut2 expression was made manually each ~200 µm. The neuron was considered as positive only if it contained at least eight dots of transcripts for the mRNA co-localized with a nucleus labeled with DAPI. The percentage of double (MuGFP/VGlut2 or MuGFP/VGAT) labeled neurons was calculated in relation to the total number of MuGFP cells that were quantified bilaterally.

## Data analysis and statistics

Analysis of the muscle activities was performed on rectified and smoothed signals (time constant of 50 ms). dEMG activity was evaluated by its burst amplitude (DIA, V) and frequency (fR, breaths/min). The DIA amplitude, fR, BP, and HR values were extracted from Spike2 at the sample rate of 1 Hz and were graphically represented as absolute values. These values were taken in three different periods: during baseline (15 s before the beginning of photoinhibition), during the 15–30 s of photoinhibition and during the recovery period (15 s post-photoinhibition). SpikeSorting analysis (Spike2 software, Cambridge Electronic Design) was also employed to evaluate to compare the temporal activity of MP during baseline and during photoinhibition. The quantification of the synchronization between MP activity and diaphragm activity during the selective photoinhibition of

7n-projecting preBötC neurons was also performed by analyzing the event-triggered average of MP activity (triggered by the rapid rise of dEMG that occurs at the beginning of inspiration) over 5–8 respiratory cycles during baseline and during the initial period of photoinhibition. The magnitude of inspiratory-related modulation of MP activity in the 150–200 ms before and after the onset of inspiration was plotted in graphs.

The function 'modulus' of Spike2 was used to extract the areas under the curve of the raw signal of dEMG, AbdEMG, and mystacial pad EMG activities. The area under the curve values and the changes in SAP, HR, and fR were extracted from three different periods: during baseline (5 respiratory cycles before the beginning of the photoinhibition), during the photoinhibition, and during the recovery (5 respiratory cycles after the end of the photoinhibition). The photoinhibition period was further divided into two periods: P1 (extracted during the 5 first respiratory cycles from the beginning of photoinhibition or, in the case where apnea was present, was collected the same extension as the 5 respiratory cycles during baseline) and P2 (5 respiratory cycles after the breathing resumption).

All sets of data were submitted to the normality test (SigmaPlot v11, CA, USA). In cases where the set of data failed to pass the normality test, the data was assessed by the nonparametric Mann-Whitney test with multiple comparisons using the Bonferroni-Dunn method (GraphPad Prism9, La Jolla, CA, USA). If the data passed the normality test, two-way ANOVA followed by Bonferroni post hoc. Statistical analysis of the period P1 was performed using either the unpaired t-test, if the set of the data passed the normality test or the nonparametric Mann-Whitney U test.

All data are described as the mean (95% confidence interval) and were graphically represented as mean ± 95% CI using GraphPad (GraphPad Prism9, La Jolla, CA, USA). Differences were considered significant at $p < 0.05$.

## Acknowledgements

The authors acknowledge the facilities and technical assistance of the Biological Optical Microscopy Platform (University of Melbourne) for the confocal microscopy images.

## Additional information

### Funding

| Funder | Grant reference number | Author |
|---|---|---|
| Early Career Research - University of Melbourne | 503275 | Mariana R Melo |
| Early Career Reasearch Transition Grant of Hypertension Australia Ltd | 830365 | Mariana R Melo |
| Australian Research Council | DP231003058 | Andrew M Allen |
| National Health and Medical Research Council | 1156727 | Andrew M Allen |

The funders had no role in study design, data collection and interpretation, or the decision to submit the work for publication.

### Author contributions

Mariana R Melo, Andrew M Allen, Conceptualization, Resources, Data curation, Software, Formal analysis, Supervision, Funding acquisition, Validation, Investigation, Visualization, Methodology, Writing – original draft, Project administration, Writing – review and editing; Alexander D Wykes, Conceptualization, Resources, Validation, Investigation, Methodology, Writing – original draft; Angela A Connelly, Conceptualization, Software, Formal analysis, Validation, Visualization, Methodology; Jaspreet K Bassi, Conceptualization, Software, Formal analysis, Validation, Investigation, Visualization, Methodology; Shane D Cheung, Resources, Software; Stuart J McDougall, Conceptualization, Investigation, Methodology, Writing – review and editing; Clément Menuet,

Conceptualization, Data curation, Validation, Investigation, Methodology, Writing – review and editing; Ross AD Bathgate, Resources, Funding acquisition, Validation, Investigation, Methodology, Writing – review and editing

### Author ORCIDs

Alexander D Wykes ⓘ https://orcid.org/0000-0001-8264-7585
Angela A Connelly ⓘ https://orcid.org/0000-0002-4547-4352
Shane D Cheung ⓘ https://orcid.org/0000-0001-8733-4756
Stuart J McDougall ⓘ https://orcid.org/0000-0002-8778-675X
Clément Menuet ⓘ http://orcid.org/0000-0002-7419-6427
Andrew M Allen ⓘ https://orcid.org/0000-0002-2183-5360

### Ethics

Experiments were conducted in accordance with the National Health and Medical Research Council of Australia's "Guidelines to promote the well-being of animals used for scientific purposes: The assessment and alleviation of pain and distress in research animals (2008)" and "Australian code for the care and use of animals for scientific purposes" and were approved by the University of Melbourne Animal Research Ethics and Biosafety Committees (ethics ID#21396).

### Decision letter and Author response

Decision letter https://doi.org/10.7554/eLife.85398.sa1
Author response https://doi.org/10.7554/eLife.85398.sa2

## Additional files

### Supplementary files

• MDAR checklist

### Data availability

All data generated or analysed during this study are included in the manuscript and supporting file; Source Data files have been provided for Figures 2, 3, 8, Figures 8 -supplement 1 and 3, and Figure 9.

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
