## [Editor Report]

This important study advances our understanding of the composition and circuit organization of the preBötzinger complex (preBötC), the brainstem region that generates the respiratory rhythm and coordinates breathing with different motor and physiological behaviors in mammals. The authors present convincing evidence supporting their conclusion that within the preBötC region, there is a subgroup of output neurons that has axonal projections to the facial motor nucleus and provides respiratory-related modulation of nasofacial muscle activity, based on technically elegant, state-of-the-art combinatorial dual viral transgenic and optogenetic approaches in rats. This work will be of interest to neuroscientists and physiologists working on the neural control of breathing and other motor systems.

---

## [Decision Letter]

**Decision letter after peer review:**

Thank you for submitting your article "Selective transduction and photoinhibition of pre-botzinger neurons that project to the facial nucleus in rats affect the nasofacial activity" for consideration by *eLife*. Your article has been reviewed by 3 peer reviewers, including Jeffrey C Smith as Reviewing Editor and Reviewer #1, and the evaluation has been overseen by Andrew King as the Senior Editor. The following individual involved in the review of your submission has agreed to reveal their identity: Muriel Thoby-Brisson (Reviewer #2).

Essential revisions:

1) As indicated by the reviewers, the authors should use traditional markers as a counterstain of the preBötC region to confirm the anatomical location. How can the authors be sure that they have transduced all preBötC neurons in the region ventral and caudal to the NAc, given that preBötC neurons are not definitively defined? Also please note that the coronal section shown in Figure supplement 1B appears to be caudal to the medullary levels containing the preBötC, and hence does not appear to illustrate distinct CTB retrogradely labeled preBötC subpopulations. Additional data addressing the above should be included.

2) The authors should quantify the expression of mCherry within GFP positive neurons as a control for the specificity of the CN7 projecting neurons by ruling out Cre-independent viral expression, as indicated by Reviewer #3. Please include this data.

3) There are cells non-selectively and selectively transduced by the vector combination in the vLRt (Figure 2—figure supplement 2C and D and Figure 3—figure supplement 1C and D), raising the possibility that some of the perturbations of MP activity might be attributable to photoinhibition of these cells, which is worth comment in the text.

4) The authors should explain why some preBötC neurons express muGFP that were not VGluT2^+^ or VGAT+ (Figure 4B and F).

5) The authors should discuss why the inhibition of MP EMG activity is only transient during sustained photoinhibition at the very high laser power employed.

6) The relationship between breathing and the change in mystacial activity in Figures 2, 3, 7, and 8 is not directly quantified. One possible measurement to show a loss of synchrony would be the cross-correlation between signals for breathing and mystacial muscle activity, as in Kurnikova 2017.

7) The authors should discuss the issues regarding alternative interpretations of the data raised by Reviewer #3.

8) The authors should address all other questions/issues raised in the major concerns of Reviewers #2 and 3. Additional experiments/data are not required here.

*Reviewer #1 (Recommendations for the authors):*

Concerns to be addressed:

1) The authors should discuss why the inhibition of MP EMG activity is only transient during sustained photoinhibition at the very high laser power employed.

2) There are cells non-selectively and selectively transduced by the vector combination in the vLRt (Figure 2—figure supplement 2C and D and Figure 3¬ figure supplement 1C and D), raising the possibility that some of the perturbations of MP activity might be attributable to photoinhibition of these cells, which is worth comment.

3) The authors should explain why some preBötC neurons express muGFP that were not VGluT2^+^ or VGAT+ (Figure 4B and F).

*Reviewer #2 (Recommendations for the authors):*

While the study is very elegant and the illustrations convincing, I have three concerns:

1. It remains unclear to me why the authors consider their neurons of interest (identified based on their axonal projections) as elements of the preBötzinger network. It might just be a problem of semantics: With the term preBötC, are we talking about a region or a network? Are we talking about a function (breathing) or an anatomical location? For instance are the neurons labelled here Dbx1-derived, as is the case for preBötC neurons involved in breathing (Bouvier et al., 2010; Gray et al., 2010)? Are they, for example, somatostatin or NK1R positive? In the present version of the manuscript, it seems that the identification of neurons as being part of the preBötC is only based on their anatomical location. Could the authors provide additional arguments in favor (or not) for these neurons to express any known features of the respiratory preBötC neurons? Or do the authors consider that the neurons of interest are located in the same region as respiratory preBötC neurons but constitute a completely different type of neurons? This should be clarified and addressed in the discussion. More generally, it actually becomes a point that should be addressed in the field, as several labs are identifying small networks in the brainstem that control distinct motor functions all more or less coordinated with breathing behavior. So are we investigating one large network composed of several intermingled small networks or are they really distinct neuronal groups? What is now the definition of the preBötC?

2. The authors provide evidence that the neurons of interest are a mixture of excitatory and inhibitory phenotypes. The photo-inhibition targets indistinctly either type. It is obvious to expect opposite effects when inhibiting inhibitory neurons versus excitatory neurons. So how do the authors explain the overall effects? Which sub-population (excitatory or inhibitory) is mainly involved in this effect? Are inhibitory and excitatory neurons involved in controlling the same orofacial muscles? Are they sending collaterals to the same brainstem nuclei? This point deserves to be developed further in the discussion (within the text line 435, page 19).

And of course, this raises the question of what would happen if the labelled neurons would be photo-activated instead of photo-inhibited. Answering this question would imply performing an additional set of experiments with different viral construction, and I don't think this could reasonably be asked of the authors, but they could at least discuss this point.

3. Sites of projection: it is surprising that the projections of preBötC neurons with the non-specific transduction do not invade more globally the hypoglossal nucleus (Figure 5). In contrast, the labelling appears strongly restricted to a very limited number of hypoglossal motoneurons. Is this just due to the choice of the image or is it really representative? Even more surprising when we compare the same image with the one obtained with selective transduction (Figure 6). Here the labelling invades a large portion of the XII nucleus. It is difficult to understand how fewer neurons could send more projections to the same target. Also, can you explain why the selective transduction does not result in the specific labelling of lateral and dorsal lateral border of the facial nucleus, as expected due to the approach used??

4. Personal curiosity: Do the authors have indications about the pattern of discharge of preBötC→7n neurons?

*Reviewer #3 (Recommendations for the authors):*

If these can be addressed, the manuscript will be more decisive.

Does inhibition of a similar number of preBötC neurons impact breathing?

While silencing many preBötC versus the subset that projects to the facial nucleus impact breathing to different magnitudes, none-the-less, both manipulations slow/stop breathing (most obvious in Figure 8). The magnitude of this effect changes upon the use of anesthesia and the type and depth. An alternative interpretation from the authors of this experiment is that inhibition of a random small number of preBötC neurons (matching the number that projects to the facial nucleus) would similarly slow breathing. The experiments conducted cannot rule this out and such a finding would re-sculpt the main message of the manuscript.

Does inhibitions of preBötC neurons that do not project to the mystacial pad prevent respiratory modulation of the nares?

The preBötC neurons labeled by retrograde tracing from the facial nucleus are presumed to directly control the breathing modulation of the nares. While this is a simple conclusion, the experiments cannot exclude that the partial slowing of breathing is indirectly changing the movement of the nares. For example, alternative premotor populations, like the nIRT or nRF (Kurnikova 2019), that receive preBötC input are those that also modulate the activity of the muscles in the nares. In such a case, the axonal targeting of CN7 would have a small effect on the modulation of the mystacial pad. This might be addressed by inhibiting the axons of these preBötC neurons within the facial nucleus.

Cross correlation of mystacial activity with breathing.

The authors conclude that the respiratory activity of the mystacial pad is lost by inhibition of the CN7 projecting preBötC neurons. While the activity of the mystacial pad muscle changes (mostly to tonic states), the relationship between breathing and the change in mystacial activity in Figures 2, 3, 7, and 8 is not directly quantified. One possible measurement to show a loss of synchrony would be the cross-correlation between breathing and mystacial muscle activity, as in Kurnikova 2017.

Other points to address

– Figure 1C: It would be helpful to use a molecular marker to denote the preBötC (like sst/Nk1R/oprm1) to demonstrate that the labeled neurons are within the anatomical location of the preBötC. In panel C, many labeled cells appear ventral to the preBötC intermingled with other medullary groups that express TH. A similar analysis of the retrogradely labeled neurons in Figure 4F would demonstrate preBötC localization.

– Figure 2 and 3: Quantification of the expression of mCherry within GFP positive neurons. This is expected to be 100% and serves as a control for claiming specificity for the CN7 projecting neurons by ruling out cre-independent viral expression.

– Figure 2 D/K vs. 3 D/K: Explanation for why mystacial pad activity increases when the entire preBötC is inhibited but the pad activity decreases when the CN7 projecting neurons are silenced. Also, as in Figure 2, the remaining figures show the mystacial pad activity increasing in a similar direction between both experimental cohorts.

---

## [Author Response]

Essential revisions:1) As indicated by the reviewers, the authors should use traditional markers as a counterstain of the preBötC region to confirm the anatomical location. How can the authors be sure that they have transduced all preBötC neurons in the region ventral and caudal to the NAc, given that preBötC neurons are not definitively defined? Also please note that the coronal section shown in Figure supplement 1B appears to be caudal to the medullary levels containing the preBötC, and hence does not appear to illustrate distinct CTB retrogradely labeled preBötC subpopulations. Additional data addressing the above should be included.

We agree with the reviewers that due to technical limitations and the lack of specific neurochemical signatures that define preBötC, we cannot claim that all preBötC neurons have been transduced by the viral vector used in this study. This statement has been modified in the revised version of the manuscript. To better characterize, anatomically and phenotypically, the neuronal population of preBötC transduced by viral injections, we also have included in the manuscript new data (Figures 4 and 5) showing the co-localization between muGFP and reeling, somatostatin, NK1R, which are typical markers used to define preBötC neurons (pages 9).

2) The authors should quantify the expression of mCherry within GFP positive neurons as a control for the specificity of the CN7 projecting neurons by ruling out Cre-independent viral expression, as indicated by Reviewer #3. Please include this data.

As suggested by the reviewer, we quantified the expression of mCherry within muGFP-positive neurons. We found that 48.5% (95% CI: 12.77 to 84.23%) of muGFP neurons also expressed mCherry. We recognize that the percentage of co-localization seems to be lower than expected. However, considering that the Cre-loxP system requires only a single molecule of Cre-recombinase enzyme (Van Duyne, 2015), it is plausible that low levels of Cre, that were not detected by immunohistochemistry, had resulted in the expression of muGFP. We believe that the results shown in Figure 2—figure supplement 2 are more important to rule out the Cre-independent viral expression, showing that the control injections of AAV-DIO-GtACR2-MuGFP into preBötC did not produce any expression of muGFP in the absence of expression of Cre (pages 9-10, lines 212-217).

3) There are cells non-selectively and selectively transduced by the vector combination in the vLRt (Figure 2—figure supplement 2C and D and Figure 3—figure supplement 1C and D), raising the possibility that some of the perturbations of MP activity might be attributable to photoinhibition of these cells, which is worth comment in the text.

We thank the Reviewer for raising this point. Whilst every attempt is made to develop injection protocols that effectively target the cell population of interest, this is next to impossible and other neurons are affected. We have added a short section to the Discussion (page 17-18, lines 399-406) to address this issue and draw the reader’s attention to additional neurons transduced in other nuclei.

4) The authors should explain why some preBötC neurons express muGFP that were not VGluT2^+^ or VGAT+ (Figure 4B and F).

We thank the reviewer for noting this, however we do not have a clear understanding of why a small proportion of the transduced neurons express neither VGluT2 nor VGAT. The simplest explanation is that it is due to technical limitations and the strict criteria employed for quantifying expression. A neuron was only considered positive for VGAT or Vglut2 if it contained at least eight transcript dots for the mRNA, and that these were co-localized with a nucleus labeled with DAPI. During the quantification process, it was common to observe MuGFP-positive neurons that expressed four or fewer dots of transcript for VGAT or VGlut2 mRNA or neurons that were not co-localized with DAPI. These neurons were not included in either of the groups MuGFP + VGlut2 or MuGFP +

VGAT. Alternative hypotheses are the existence of other vesicle transporters that haven’t been identified yet, or that these neurons utilize transmitters other than glutamate, GABA, and glycine. We do not have further insight into this neurochemistry.

5) The authors should discuss why the inhibition of MP EMG activity is only transient during sustained photoinhibition at the very high laser power employed.

We thank the reviewer for their helpful comments and overall support for this manuscript. We have observed that photoinhibition of preBötC transiently inhibits breathing, blood pressure (Menuet et al., 2020), and now MP activity. In contrast, the bradycardic effect of photoinhibition, induced by vagal activation, is retained for the length of laser delivery. In unpublished data of extracellular recordings of preBötC neurons, a similar phenomenon is observed, with continuous inhibition of some, but only transient inhibition of other neurons. We are attempting to understand this mechanism. One hypothesis is that the maintenance of the chloride gradient is different between different neuronal populations. Clearly, we are interested in whether this difference provides insight into the different neuronal groups and plan to explore this further. We have added a section to the Discussion to address this issue (page 15, lines 330-337).

6) The relationship between breathing and the change in mystacial activity in Figures 2, 3, 7, and 8 is not directly quantified. One possible measurement to show a loss of synchrony would be the cross-correlation between signals for breathing and mystacial muscle activity, as in Kurnikova 2017.

In consideration of the reviewer’s helpful suggestion, we have quantified the relationship between breathing and mystacial pad activity during the selective photoinhibition of 7n projecting preBötC neurons. We evaluated the synchronization between MP activity and diaphragm activity by analyzing the event-triggered average (ETA) of MP activity (triggered by the rapid rise of dEMG that occurs at the beginning of inspiration). The comparison was made over 5-8 respiratory cycles during baseline and during the initial period of photoinhibition. We haven’t applied this analysis for the nonselective transduction of preBötC neurons, as photoinhibition of preBötC neurons induced apnea, thus the ETA analysis could not be done. The description of these results (pages 8-9, 11-13), as well as their corresponding new figures (Figure 3E, Figure 8 —figure supplement 3), are included in the revised version of the manuscript.

Selective inhibition of preBötC neurons projecting to 7n in urethane anesthetized rats

In urethane-anesthetized rats, we found that photoinhibition of preBötC→7neurons resulted in the loss of respiratory modulation of MP activity (3.9 mV [95% CI: 1 to 6.8 mV] vs. baseline 26.6 mV [95% CI: 9.4 to 43.8 mV], p=0.02), whilst producing a minimal reduction of dEMG (23.8 mV [95% CI: 9.9 to 37.5 mV] vs. 29.1 mV [95% CI: 11.2 to 47 mV], p=0.04) (Figure 3E) (pages 8-9, lines 187-193).

Selective inhibition of preBötC neurons projecting to 7n in conscious rats

We also quantified the respiratory-related activity of MP in rats under three different conditions: ketamine/medetomidine anesthesia, during the very early stage of regaining consciousness, a few min after the anesthesia was reversed with atipamezole, and 1-2 h after the atipamezole injection, when the animal had recovered from anesthesia and was completely awake and freely behaving. The results show that both activity and respiratory modulation of MP are dependent on the rat’s state/behaviour.

– Surgical ketamine/medetomidine anesthesia

Under ketamine/medetomidine anesthesia, the MP activity is minimal and nonrespiratory modulated (Figure 8—figure supplement 3A). Photoinhibition of preBötC→7n did not produce any changes in MP activity (page 11, lines 254-257).

– Early recovery phase.

During the very early stage of regaining consciousness, the inspiratory modulation of the mystacial pad activity resumed in all animals. Despite this, two different responses were observed on MP respiratory-related activity during photoinhibition of preBötC→7n. We observed that in one sub-cohort (n=3), photoinhibition flattened the inspiratory modulation of MP (1 mV [95% CI: 0.09 to 1.9 mV] vs. baseline: 6.8 mV [95% CI: 6.4 to 7.8 mV], p=0.01) even though it did not affect the dEMG amplitude (40.3 mV [95% CI: 25 to 58 mV] vs. baseline: 32.3 mV [95% CI: 20.2 to 52.8 mV], p=0.25). Interestingly, an increase in respiratory-related MP activity was observed in the second sub-cohort (n=2)two rats tested in this experiment (rat 1: 9.6 mV vs. baseline: 3.3 mV; rat 2: 17.6 mV vs. baseline: 8 mV). Again, dEMG was not affected by photoinhibition (rat 1: 51 mV vs. baseline: 56 mV; rat 2: 43.9 mV vs. baseline: 41.7 mV) (page 12, lines 271-276).

– Conscious phase.

Variable responses on the respiratory modulation of MP to photoinhibition of preBötC→7n were observed in conscious rats. We found out that, in 3 out of 5 rats, the ETA analysis did not show respiratory modulation of MP activity at either baseline or during photoinhibition. Given the high respiratory frequency, we speculate photoinhibition was performed during sniffing/whisking behaviour. Intriguingly, we also found that the same two rats that presented inspiratory-related MP activity during the early recovery phase from ket/med anesthesia, had expiratory-related activity when they were fully conscious. Moreover, while the selective photoinhibition of preBötC→7n increased the magnitude of the expiratory activity of MP in one of these rats, it decreased in the other one. These distinct responses are shown in Figure 8 – supplement 2C. It is important to note that despite these different effects in the respiratory modulation of MP, the overall activity of MP, measured as AUC, increased in all rats tested in this experimental protocol (Figure 8T, Figure 9T) (pages 12-13, lines 278-286).

Our results show that selective photoinhibition of preBötC→7n significantly changes the MP activity without affecting the inspiratory rhythm generation regardless of the rat’s state. We don’t know the reasons why the responses were heterogeneous among the rats. It is possible that the MP activity recorded in animals regaining consciousness or when they were fully conscious reflected the activity of multiple inspiratory and expiratory-related muscles, given the difficulty of dissecting the rodent snout (Haidarliu et al., 2012), the large number of distinct muscles that compose the mystacial pad (Haidarliu et al., 2010) and the substantial size of the electrode's suture pads. Another possibility is that the responses induced by photoinhibition could be related to the animal’s behaviour, such as resting, sniffing, and whisking. Unfortunately, in our experiment, photoinhibition was indiscriminately applied during any behaviour (pages 16-17, lines 370-386).

7) The authors should discuss the issues regarding alternative interpretations of the data raised by Reviewer #3.

We have addressed the issues regarding alternative interpretations of the data raised by Reviewer #3 in the responses to their questions 1 and 2, and have included this in the revised version of the manuscript (page 19, lines 428-443).

8) The authors should address all other questions/issues raised in the major concerns of Reviewers #2 and 3. Additional experiments/data are not required here.

We have addressed all the questions/issues raised by Reviewers #2 and 3. Additional discussion have been added in the revised version of the manuscript (page 19, lines 428-443).

Reviewer #1 (Recommendations for the authors):Concerns to be addressed:1) The authors should discuss why the inhibition of MP EMG activity is only transient during sustained photoinhibition at the very high laser power employed.

We thank the reviewer for their helpful comments and overall support for this manuscript. We have observed that photoinhibition of preBötC transiently inhibits breathing, blood pressure (Menuet et al., 2020), and now MP activity. In contrast, the bradycardic effect of photoinhibition, induced by vagal activation, is retained for the length of laser delivery. In unpublished data of extracellular recordings of preBötC neurons, a similar phenomenon is observed, with continuous inhibition of some, but only transient inhibition of other neurons. We are attempting to understand this mechanism. One hypothesis is that the maintenance of the chloride gradient is different between different neuronal populations. Clearly, we are interested in whether this difference provides insight into the different neuronal groups and plan to explore this further. We have added a section to the Discussion to address this issue (page 15, lines 330-337).

2) There are cells non-selectively and selectively transduced by the vector combination in the vLRt (Figure 2—figure supplement 2C and D and Figure 3¬ figure supplement 1C and D), raising the possibility that some of the perturbations of MP activity might be attributable to photoinhibition of these cells, which is worth comment.

We thank the Reviewer for raising this point. Whilst every attempt is made to develop injection protocols that effectively target the cell population of interest, this is next to impossible, and other neurons are affected. We have added a short section to the Discussion (pages 17-18, lines 399-406) to address this issue and draw the reader’s attention to additional neurons transduced in other nuclei.

3) The authors should explain why some preBötC neurons express muGFP that were not VGluT2^+^ or VGAT+ (Figure 4B and F).

We thank the reviewer for noting this, however, we do not have a clear understanding of why a small proportion of the transduced neurons express neither VGluT2 nor VGAT. The simplest explanation is that it is due to technical limitations and the strict criteria employed for quantifying expression. A neuron was only considered positive for VGAT or Vglut2 if it contained at least eight transcript dots for the mRNA, and that these were co-localized with a nucleus labeled with DAPI. During the quantification process, it was common to observe MuGFP-positive neurons that expressed four or fewer dots of transcript for VGAT or VGlut2 mRNA or neurons that were not co-localized with DAPI. These neurons were not included in either of the groups MuGFP + VGlut2 or MuGFP + VGAT. Alternative hypotheses are the existence of other vesicle transporters that haven’t been identified yet, or that these neurons utilize transmitters other than glutamate, GABA, and glycine. We do not have further insight into this neurochemistry.

Reviewer #2 (Recommendations for the authors):While the study is very elegant and the illustrations convincing, I have three concerns:1. It remains unclear to me why the authors consider their neurons of interest (identified based on their axonal projections) as elements of the preBötzinger network. It might just be a problem of semantics: With the term preBötC, are we talking about a region or a network? Are we talking about a function (breathing) or an anatomical location? For instance are the neurons labelled here Dbx1-derived, as is the case for preBötC neurons involved in breathing (Bouvier et al., 2010; Gray et al., 2010)? Are they, for example, somatostatin or NK1R positive? In the present version of the manuscript, it seems that the identification of neurons as being part of the preBötC is only based on their anatomical location. Could the authors provide additional arguments in favor (or not) for these neurons to express any known features of the respiratory preBötC neurons? Or do the authors consider that the neurons of interest are located in the same region as respiratory preBötC neurons but constitute a completely different type of neurons? This should be clarified and addressed in the discussion. More generally, it actually becomes a point that should be addressed in the field, as several labs are identifying small networks in the brainstem that control distinct motor functions all more or less coordinated with breathing behavior. So are we investigating one large network composed of several intermingled small networks or are they really distinct neuronal groups? What is now the definition of the preBötC?

We thank the reviewer for their insight and constructive comments. This is an interesting comment and the reviewer’s concerns regarding the definition of the preBötC resonate with us. Our interpretation of the large body of work from other laboratories is that the preBötC is a complex, as opposed to a nucleus, that occupies an anatomical location. Within that location are neurons that generate inspiratory rhythm. These neurons have a particular neurochemistry (Dbx-1; VGluT2; SST; NK1R) but none of these markers are exclusive identifiers. Neurons with these neurochemistries project from the preBötC to other brain regions and are presumably involved in functions other than inspiratory rhythm generation.

We have transduced neurons in the preBötC on the basis of anatomical coordinates, relative to obex, and the strong inspiratory activity recorded at the site of injection with the injection pipette. In our initial study (Menuet et al., 2020), we demonstrated that these injections transduced neurons caudal to the parvalbumin-expressing neurons of the Bötzinger nucleus, ventral to the subcompact formation of the nucleus ambiguus and predominantly dorsal to, but intermingled with, the catecholaminergic neurons of the ventrolateral medulla. We also observed that the caudal extent of the transduction had some overlap with the very rostral extent of the parvalbumin-expressing neurons of the rVRG. The distribution of transduced neurons in this study is near identical.

We did not provide evidence in this study of the neurochemical phenotype of the transduced neurons, other than primary ionotropic neurotransmitter, as we did not consider any particular marker definitive. However, in response to the reviewer’s request, we have now included evidence that different groups of transduced neurons express reelin, SST, or NK1R, thus marking them a neurons traditionally ascribed to the preBötC (new Figure 5EG). We also show that some non-selectively transduced preBötC neurons also express NK1R, SST, reelin, or both (new figure 4E-G).

We do not consider that our data is sufficient to adequately address the question of the exact nature of the neurons we have studied, relative to rhythm-generating neurons. As indicated by Reviewer 3, there are several potential explanations for our data. Our working hypothesis is that the facial nucleus projecting preBötC neurons are transmitting respiratory rhythm, rather than being responsible for generating it. The small effect of photoinhibiting these neurons on inspiratory activity is due to collaterals. However, it is possible, as suggested by Reviewer 3, that the neurons projecting to the facial nucleus are a sub-group of the rhythm-generating neurons with a collateral to the facial nucleus – collateral being a potentially confusing term. Without being over-speculative, we have attempted to briefly address this issue in the Discussion (page 19, lines 428-443), but know that a different experimental dataset is required to directly answer the Reviewer’s interesting question.

2. The authors provide evidence that the neurons of interest are a mixture of excitatory and inhibitory phenotypes. The photo-inhibition targets indistinctly either type. It is obvious to expect opposite effects when inhibiting inhibitory neurons versus excitatory neurons. So how do the authors explain the overall effects? Which sub-population (excitatory or inhibitory) is mainly involved in this effect? Are inhibitory and excitatory neurons involved in controlling the same orofacial muscles? Are they sending collaterals to the same brainstem nuclei? This point deserves to be developed further in the discussion (within the text line 435, page 19).And of course, this raises the question of what would happen if the labelled neurons would be photo-activated instead of photo-inhibited. Answering this question would imply performing an additional set of experiments with different viral construction, and I don't think this could reasonably be asked of the authors, but they could at least discuss this point.

The reviewer raises another very interesting question which we can only partially address with our data. Previous studies, e.g. Yang and Feldman, J Comp Neurol 526: 13891402 (2018) (Yang and Feldman, 2018) have clearly demonstrated that the preBötC sends parallel inhibitory and excitatory projections widely throughout the neuraxis. Our data confirm this in relation to projections to the facial nucleus. An interesting study would be to examine the effect of photoinhibiting one sub-group, either excitatory or inhibitory, however, that is not within our genetic targeting tool-kit at this stage. Our working hypothesis is that the influence of the preBötC on orofacial motor activity will be statedependent, with a predominance of excitatory or inhibitory influence under different conditions. Whilst potentially interesting, we are not sure that photoactivation would provide a particularly informative result as the coordinated, coincident activation of all neurons of a particular phenotype is unlikely to be physiological. For this reason, we favour photoinhibition and predict that definitive answers will arise from more selective transduction and inhibition under different conditions. Without being over-speculative, we have attempted to briefly address this issue in the Discussion (pages 20-21, lines 467-477).

3. Sites of projection: it is surprising that the projections of preBötC neurons with the non-specific transduction do not invade more globally the hypoglossal nucleus (Figure 5). In contrast, the labelling appears strongly restricted to a very limited number of hypoglossal motoneurons. Is this just due to the choice of the image or is it really representative? Even more surprising when we compare the same image with the one obtained with selective transduction (Figure 6). Here the labelling invades a large portion of the XII nucleus. It is difficult to understand how fewer neurons could send more projections to the same target. Also, can you explain why the selective transduction does not result in the specific labelling of lateral and dorsal lateral border of the facial nucleus, as expected due to the approach used??

We thank the reviewer for noting this. We made a technical error in adjusting the image for publication and in the process of reducing the background fluorescence, reduced detectability of some axonal projections. We have re-adjusted the same image to correctly depict the level of axon terminal input. In both cases, the axonal projections from preBötC invades the whole hypoglossal nucleus.

4. Personal curiosity: Do the authors have indications about the pattern of discharge of preBötC→7n neurons?

This is an interesting question, but we do not have further data in relation to that at this stage.

Reviewer #3 (Recommendations for the authors):If these can be addressed, the manuscript will be more decisive.Does inhibition of a similar number of preBötC neurons impact breathing?While silencing many preBötC versus the subset that projects to the facial nucleus impact breathing to different magnitudes, none-the-less, both manipulations slow/stop breathing (most obvious in Figure 8). The magnitude of this effect changes upon the use of anesthesia and the type and depth. An alternative interpretation from the authors of this experiment is that inhibition of a random small number of preBötC neurons (matching the number that projects to the facial nucleus) would similarly slow breathing. The experiments conducted cannot rule this out and such a finding would re-sculpt the main message of the manuscript.

We accept the reviewer’s comment and potential alternative interpretation of the data. The hypothesis would be that sub-groups of rhythm-generating/transmitting preBötC neurons have collaterals to another specific region. Selective transduction, and photoinhibition, of all of these sub-groups would result in total inhibition of inspiratory rhythm generation. We have added a short section to the Discussion to address this issue (page 19, lines 428-443).

Does inhibitions of preBötC neurons that do not project to the mystacial pad prevent respiratory modulation of the nares?The preBötC neurons labeled by retrograde tracing from the facial nucleus are presumed to directly control the breathing modulation of the nares. While this is a simple conclusion, the experiments cannot exclude that the partial slowing of breathing is indirectly changing the movement of the nares. For example, alternative premotor populations, like the nIRT or nRF (Kurnikova 2019), that receive preBötC input are those that also modulate the activity of the muscles in the nares. In such a case, the axonal targeting of CN7 would have a small effect on the modulation of the mystacial pad. This might be addressed by inhibiting the axons of these preBötC neurons within the facial nucleus.

Again we thank the reviewer for their insight and helpful alternative explanations for our data. We contemplated placing the optical fibres in the facial nucleus to see what effect this might have. Our data, including unpublished extracellular recordings of preBötC neurons, demonstrate inhibition when light is directed to the somata. However, through the observations of others, mostly unpublished, we are cautious about using GtACR2 to ‘inhibit’ axonal terminals. Possibly due to differences in chloride gradients in different neuronal compartments, others have reported increased synaptic activity in response to light-induced activation of GtACR2 (Messier et al., 2018). The reviewer’s suggestion might be better tested using the more recently described light-activated K^+^ channels, or the Gi coupled eOPN3. We have added a short section to the Discussion to address the issue raised by the Reviewer’s comment (page 19, lines 428-443).

Cross correlation of mystacial activity with breathing.The authors conclude that the respiratory activity of the mystacial pad is lost by inhibition of the CN7 projecting preBötC neurons. While the activity of the mystacial pad muscle changes (mostly to tonic states), the relationship between breathing and the change in mystacial activity in Figures 2, 3, 7, and 8 is not directly quantified. One possible measurement to show a loss of synchrony would be the cross-correlation between breathing and mystacial muscle activity, as in Kurnikova 2017.

In consideration of the reviewer’s helpful suggestion, we have quantified the relationship between breathing and mystacial pad activity during the selective photoinhibition of 7n projecting preBötC neurons. We evaluated the synchronization between MP activity and diaphragm activity by analyzing the event-triggered average (ETA) of MP activity (triggered by the rapid rise of dEMG that occurs at the beginning of inspiration). The comparison was made over 5-8 respiratory cycles during baseline and during the initial period of photoinhibition. We haven’t applied this analysis for the nonselective transduction of preBötC neurons, as photoinhibition of preBötC neurons induced apnea, thus the ETA analysis could not be done. The description of these results (pages 8-9, 11-13), as well as their corresponding new figures (Figure 3E, Figure 8 —figure supplement 3), are included in the revised version of the manuscript.

Selective inhibition of preBötC neurons projecting to 7n in urethane anesthetized rats

In urethane-anesthetized rats, we found that photoinhibition of preBötC→7neurons resulted in the loss of respiratory modulation of MP activity (3.9 mV [95% CI: 1 to 6.8 mV] vs. baseline 26.6 mV [95% CI: 9.4 to 43.8 mV], p=0.02), whilst producing a minimal reduction of dEMG (23.8 mV [95% CI: 9.9 to 37.5 mV] vs. 29.1 mV [95% CI: 11.2 to 47 mV], p=0.04) (Figure 3E) (pages 8-9, lines 187-193).

Selective inhibition of preBötC neurons projecting to 7n in conscious rats

We also quantified the respiratory-related activity of MP in rats under three different conditions: ketamine/medetomidine anesthesia, during the very early stage of regaining consciousness, a few min after the anesthesia was reversed with atipamezole, and 1-2 h after the atipamezole injection, when the animal had recovered from anesthesia and was completely awake and freely behaving. The results show that both activity and respiratory modulation of MP are dependent on the rat’s state/behaviour.

– Surgical ketamine/medetomidine anesthesia

Under ketamine/medetomidine anesthesia, the MP activity is minimal and nonrespiratory modulated (Figure 8—figure supplement 3A). Photoinhibition of preBötC→7n did not produce any changes in MP activity (page 11, lines 254-257).

– Early recovery phase.

During the very early stage of regaining consciousness, the inspiratory modulation of the mystacial pad activity resumed in all animals. Despite this, two different responses were observed on MP respiratory-related activity during photoinhibition of preBötC→7n. We observed that in one sub-cohort (n=3), photoinhibition flattened the inspiratory modulation of MP (1 mV [95% CI: 0.09 to 1.9 mV] vs. baseline: 6.8 mV [95% CI: 6.4 to 7.8 mV], p=0.01) even though it did not affect the dEMG amplitude (40.3 mV [95% CI: 25 to 58 mV] vs. baseline: 32.3 mV [95% CI: 20.2 to 52.8 mV], p=0.25). Interestingly, an increase in respiratory-related MP activity was observed in the second sub-cohort (n=2)two rats tested in this experiment (rat 1: 9.6 mV vs. baseline: 3.3 mV; rat 2: 17.6 mV vs. baseline: 8 mV). Again, dEMG was not affected by photoinhibition (rat 1: 51 mV vs. baseline: 56 mV; rat 2: 43.9 mV vs. baseline: 41.7 mV) (page 12, lines 271-276).

– Conscious phase.

Variable responses on the respiratory modulation of MP to photoinhibition of preBötC→7n were observed in conscious rats. We found out that, in 3 out of 5 rats, the ETA analysis did not show respiratory modulation of MP activity at either baseline or during photoinhibition. Given the high respiratory frequency, we speculate photoinhibition was performed during sniffing/whisking behaviour. Intriguingly, we also found that the same two rats that presented inspiratory-related MP activity during the early recovery phase from ket/med anesthesia, had expiratory-related activity when they were fully conscious. Moreover, while the selective photoinhibition of preBötC→7n increased the magnitude of the expiratory activity of MP in one of these rats, it decreased in the other one. These distinct responses are shown in Figure 8 – supplement 2C. It is important to note that despite these different effects in the respiratory modulation of MP, the overall activity of MP, measured as AUC, increased in all rats tested in this experimental protocol (Figure 8T, Figure 9T) (pages 12-13, lines 278-286).

Our results show that selective photoinhibition of preBötC→7n significantly changes the MP activity without affecting the inspiratory rhythm generation regardless of the rat’s state. We don’t know the reasons why the responses were heterogeneous among the rats. It is possible that the MP activity recorded in animals regaining consciousness or when they were fully conscious reflected the activity of multiple inspiratory and expiratory-related muscles, given the difficulty of dissecting the rodent snout (Haidarliu et al., 2012), the large number of distinct muscles that compose the mystacial pad (Haidarliu et al., 2010) and the substantial size of the electrode's suture pads. Another possibility is that the responses induced by photoinhibition could be related to the animal’s behaviour, such as resting, sniffing, and whisking. Unfortunately, in our experiment, photoinhibition was indiscriminately applied during any behaviour (pages 16-17, lines 370-386).

Other points to address– Figure 1C: It would be helpful to use a molecular marker to denote the preBötC (like sst/Nk1R/oprm1) to demonstrate that the labeled neurons are within the anatomical location of the preBötC. In panel C, many labeled cells appear ventral to the preBötC intermingled with other medullary groups that express TH. A similar analysis of the retrogradely labeled neurons in Figure 4F would demonstrate preBötC localization.

In consideration of the reviewer's suggestion, we have now characterized, anatomically and phenotypically, the neuronal population of preBötC transduced by viral injections. These results are shown in the new Figure 5 of the revised version of the manuscript. Briefly, the new data show the expression of muGFP in different groups of neurons expressing reelin, somatostatin, or NK1R – typical markers used to define preBötC neurons (Figure 5E-G).

– Figure 2 and 3: Quantification of the expression of mCherry within GFP positive neurons. This is expected to be 100% and serves as a control for claiming specificity for the CN7 projecting neurons by ruling out cre-independent viral expression.

As suggested by the reviewer, we quantified the expression of mCherry within muGFP-positive neurons. We found that 48.5% (95% CI: 12.77 to 84.23%) of muGFP neurons also expressed mCherry. We recognize that the percentage of co-localization seems to be lower than expected. However, considering that the Cre-loxP system requires only a single molecule of Cre-recombinase enzyme (Van Duyne, 2015), it is plausible that low levels of Cre, that were not detected by immunohistochemistry, had resulted in the expression of muGFP. We believe that the results shown in Figure 2—figure supplement 2 are more important to rule out the Cre-independent viral expression, showing that the control injections of AAV-DIO-GtACR2-MuGFP into preBötC did not produce any expression of muGFP in the absence of expression of Cre (pages 9-10, lines 212-217).

– Figure 2 D/K vs. 3 D/K: Explanation for why mystacial pad activity increases when the entire preBötC is inhibited but the pad activity decreases when the CN7 projecting neurons are silenced. Also, as in Figure 2, the remaining figures show the mystacial pad activity increasing in a similar direction between both experimental cohorts.

We thank the reviewer for raising this important question. We don’t know why silencing preBotC→7n decreases mystacial pad activity in urethane-anesthetized rats, while the opposite response is seen when the entire preBötC is inhibited. One possible explanation might related to the population of neurons transduced – whilst a similar proportion of inhibitory and excitatory, as well as SST+, reelin+, and NK1R+ neurons were transduced by both approaches, the selective approach resulted in a much smaller population of transduced neurons with a more restricted anatomical location. Another explanation might be statedependent activation of different neuronal populations. Interestingly, mystacial pad activity increases with the photoinhibition of preBötC→7n neurons in conscious rats. Unfortunately, the experiments necessary to unravel the mechanisms that might underlie state-dependent effects are beyond this study. However, based on previous studies that reported opposite effects with interventions made in anesthetised versus conscious animals (Machado and Bonagamba, 1992; Wenker et al., 2017; Marina et al., 2011; Chiou et al., 2000), a possible explanation for our results is that under urethane, the glutamatergic preBötC→7n neurons play a major role in regulating the respiration-related mystacial pad activity, whilst the inhibitory neurons are more active in the conscious state.

References

Chitravanshi VC, Sapru HN. 1999. Phrenic nerve responses to chemical stimulation of the subregions of ventral medullary respiratory neuronal group in the rat. Brain Res 821:443– 460. doi:10.1016/S0006-8993(99)01139-7

Haidarliu S, Golomb D, Kleinfeld D, Ahissar E. 2012. Dorsorostral Snout Muscles in the Rat Subserve Coordinated Movement for Whisking and Sniffing. Anat Rec (Hoboken) 295:1181–1191. doi:10.1002/ar.22501

Haidarliu S, Simony E, Golomb D, Ahissar E. 2010. Muscle Architecture in the Mystacial Pad of the Rat. Anat Rec (Hoboken) 293:1192–1206. doi:10.1002/ar.21156

Machado BH, Bonagamba LGH. 1992. Microinjection of l-glutamate into the nucleus tractus solitarii increases arterial pressure in conscious rats. Brain Res 576:131–138. doi:10.1016/0006-8993(92)90618-J

Menuet C, Connelly AA, Bassi JK, Melo MR, Le S, Kamar J, Kumar NN, McDougall SJ, McMullan S, Allen AM. 2020. Prebötzinger complex neurons drive respiratory modulation of blood pressure and heart rate. *eLife* 9:e57288. doi:10.7554/*eLife*.57288

Messier JE, Chen H, Cai Z-L, Xue M. 2018. Targeting light-gated chloride channels to neuronal somatodendritic domain reduces their excitatory effect in the axon. *eLife* 7. doi:10.7554/*eLife*.38506

Van Duyne GD. 2015. Cre Recombinase. Microbiol Spectr 3.MDNA3-0014-2014. doi:10.1128/microbiolspec.MDNA3-0014-2014

Yang CF, Feldman JL. 2018. Efferent projections of excitatory and inhibitory preBötzinger Complex neurons. J Comp Neurol 526:1389–1402. doi:10.1002/cne.24415